# Evaluation of hydrological models on small mountainous catchments: impact of the meteorological forcings

Guillaume Evin[1], Matthieu Le Lay[2], Catherine Fouchier[3], David Penot[2], Francois Colleoni[3], Alexandre Mas[4], Pierre-André Garambois[3], and Olivier Laurantin[5]

[1]Univ. Grenoble Alpes, INRAE, CNRS, IRD, Grenoble INP, IGE, 38000 Grenoble, France
[2]EDF-DTG, Grenoble, France
[3]INRAE, Aix Marseille Univ – RECOVER, Aix-en-Provence, France
[4]Univ. Grenoble Alpes, Grenoble INP, CNRS, IRD, IGE, Grenoble, France
[5]DSO, Météo-France, Toulouse, France

**Correspondence:** Guillaume Evin (guillaume.evin@inrae.fr)

**Abstract.** Hydrological modelling of small mountainous catchments is particularly challenging because of the high spatio-temporal resolution required for the meteorological forcings. In-situ measurements of precipitation are typically scarce in these remote areas, particularly at high elevations. Precipitation reanalyses propose different alternative forcings for the simulation of streamflow using hydrological models. In this paper, we evaluate the performances of two hydrological models representing some of the key processes for small mountainous catchments ($< 300$ km$^2$), using different meteorological products with a fine spatial and temporal resolution. The evaluation is performed on 55 small catchments of the Northern French Alps. While the simulated streamflows are adequately reproduced for most of the configurations, these evaluations emphasize the added value of radar measurements, in particular for the reproduction of flood events. However, these better performances are only obtained because the hydrological models correct the underestimations of accumulated amounts (e.g. annual) from the radar data in high-elevation areas.

## 1 Introduction

Hydrological modelling of small mountainous catchments is particularly challenging for many reasons. These catchments exhibit very quick hydrological responses due to their high slopes, which require a fine temporal representation of meteorological forcings. Considering the small size of the target catchments, a high spatial resolution for precipitation is also needed, typically down to 1 km$^2$, in order to catch the spatial variability of local/intense precipitation events (Terink et al., 2018; Zhu et al., 2018; Cristiano et al., 2019). However, precipitation measurements in high-elevation areas suffer from many limitations. Weather stations are difficult to maintain in remote areas, and cover mostly low-elevation areas (Gottardi et al., 2012), while the highest intensities and annual amounts are often reached along the crests. Radar measurements suffer from beam blockage in mountainous areas (see Section 2.4 in Villarini and Krajewski, 2010) and have limited spatial coverage. Many different types of hydrometeors are also measured (light rain, heavy rain, melting snow, ice particles, etc.) which are not easily related to ground precipitation measurements (Germann et al., 2006; Khanal et al., 2019). In addition, in mountainous areas, the representation

of hydrological processes related to snow (sometimes ice) is mandatory. Finally, streamflow is particularly hard to monitor in these areas, due to the important volumes of solid transport and the very quick rise of water levels during flood events. As a consequence, long-time series of streamflow measurements are rare, which hampers hydrological applications.

Conceptual models have been extensively used to represent the key hydrological processes of small mountainous catchments. In the Alps, they have been used for the assessment of the debris flow generation processes (Simoni et al., 2020), or for climate change impacts evaluation (Aili et al., 2019). These hydrological models usually represent key hydrological processes with a flexible modelling framework and a variable number of modules, e.g. snow modules (Valéry et al., 2014), ice modules (Viviroli et al., 2009) or more complex representations of the cryosphere (Mosier et al., 2016). In France, these conceptual models have

been applied to large sets of catchments (Velázquez et al., 2010; Valéry et al., 2014; Lobligeois, 2014; Garavaglia et al., 2017; de Lavenne et al., 2019). These studies usually consider a large range of catchment size, typically between 10 km$^2$ and 10,000 km$^2$, covering partially the Northern French Alps, i.e. the region considered in this study.

Conceptual models are usually calibrated using observed meteorological forcings and streamflow. The temporal and spatial resolutions of gauged rainfall data are known to be critical for hydrological modeling (Emmanuel et al., 2017; Zeng et al.,

2018; Huang et al., 2019), particularly for small catchments (Terink et al., 2018; Hohmann et al., 2021). As an alternative, many meteorological reanalyses propose long archives of meteorological variables on a regular grid and at a relatively fine temporal resolution (e.g. hourly, daily). At the planetary scale, different products are based on numerical models of the atmosphere and land surface. Some examples are MERRA (Rienecker et al., 2011) and ERA-Interim (Dee et al., 2011) and their respective land-surface counterparts MERRA-Land (Reichle et al., 2011) and ERA-Interim/Land (Balsamo et al., 2015). They mostly

assimilate remote sensing observations. It can be viewed as an advantage considering the inhomogeneous coverage of weather stations delivering in-situ observations, in time and space, which results in temporal and spatial inconsistencies if they are integrated. Recently, a significant upgrade of ERA-Interim led to the ERA5-Land reanalysis (Muñoz-Sabater et al., 2021) available at a 9 km resolution and an hourly time scale. Alternatively, at a country or regional scale, different reanalyses are produced at a finer spatial resolution and assimilate mostly ground measurements and apply advanced interpolation techniques,

see, e.g. Vidal et al. (2010); Gottardi et al. (2012); Devers et al. (2021) in France, Frei and Schär (1998) in Switzerland or Frei and Isotta (2019) in the Alps. Finally, in the last decade, weather radars have been extensively exploited in order to provide information about precipitation at a fine temporal (hourly or sub-hourly) and spatial (1 km$^2$) resolution. In France, the French meteorological office Météo-France has delivered several composite products that correct the areal precipitation amounts obtained from the reflectivity alone with the precipitation amounts measured by the gauges (Champeaux et al., 2009).

While these different products have all been designed for land-surface applications, including hydrological applications, they have different strengths and weaknesses that impact their capacity to simulate streamflows, especially for small mountainous catchments (Villarini and Krajewski, 2010).

This paper aims to evaluate the performances of different combinations of hydrological models and precipitation reanalysis for the reproduction of streamflow in small mountainous catchments. This evaluation is carried out on 55 catchments located

in the Northern French Alps. Two hydrological models representing some of the key processes for small mountainous catchments are tested. Four different meteorological products with fine spatial and temporal resolutions are used as inputs of these

hydrological models. These different reanalyses assimilate different data information and provide different spatial/temporal resolutions. The different hydrological evaluations shown in this paper will show how the different features of the meteorological forcings impact streamflow simulations. Unlike many studies considering a large number of catchments in France, this study focuses on a selection of small mountainous catchments (less than 300 km$^2$) subject to orographic effects favoring very intense precipitation amounts and fast hydrologic concentration potentially leading to important damages (Creutin et al., 2022). A primary objective of this study is to evaluate the added value of radar information for hydrological modelling of small mountainous catchments, using the set of 55 catchments considered, compared to precipitation reanalysis based only on satellite or gauge measurements. A second objective is to assess the reliability of simulated streamflow for this type of catchment, with an emphasis on flood characteristics. The performances and limitations of these hydrological simulations are important to describe in the context of flood risk management. Section 2 presents the study area, the catchments, the meteorological and streamflow data, and the hydrological models considered in this study. In section 3, the properties of the mean areal precipitation values obtained with the different precipitation reanalysis are then compared. Section 4 evaluates the performances of the different meteo-hydrological frameworks. Section 5 provides an extensive discussion of different key points. Section 6 concludes.

## 2   Study area, meteorological data and hydrological models

### 2.1   Study area and catchments

Figure 1 shows the location of the 55 catchments considered in this study. They cover a region in the Northern French Alps delimited by the contours of French administrative entities (so-called "Département" of Drôme, Isère, Savoie, Haute-Savoie, and Hautes-Alpes). These catchments have been selected according to their size (area between 10 km$^2$ and 300 km$^2$) and the availability and quality of streamflow measurements (at least 10 years in the period 1997-2017). Streamflow data have been obtained from the French national database Banque Hydro (http://hydro.eaufrance.fr/, last date of access: 10 October 2022).

By construction, most of these catchments are small (75% are smaller than 150 km$^2$) and typical of mountainous catchments, with 25% and 75% quantiles for the median elevations equal to 580 m and 1450 m, respectively, 1380 m and 2330 m for the maximum elevations, and 9% and 25% for the mean slopes (see Table 1 for the complete list of characteristics). At the exception of a few catchments located in plains in the West of the domain, this set of catchments concerns small and steep torrential catchments largely influenced by the snow conditions during winter and spring (roughly from December to May), and for four of them by glacial contribution.

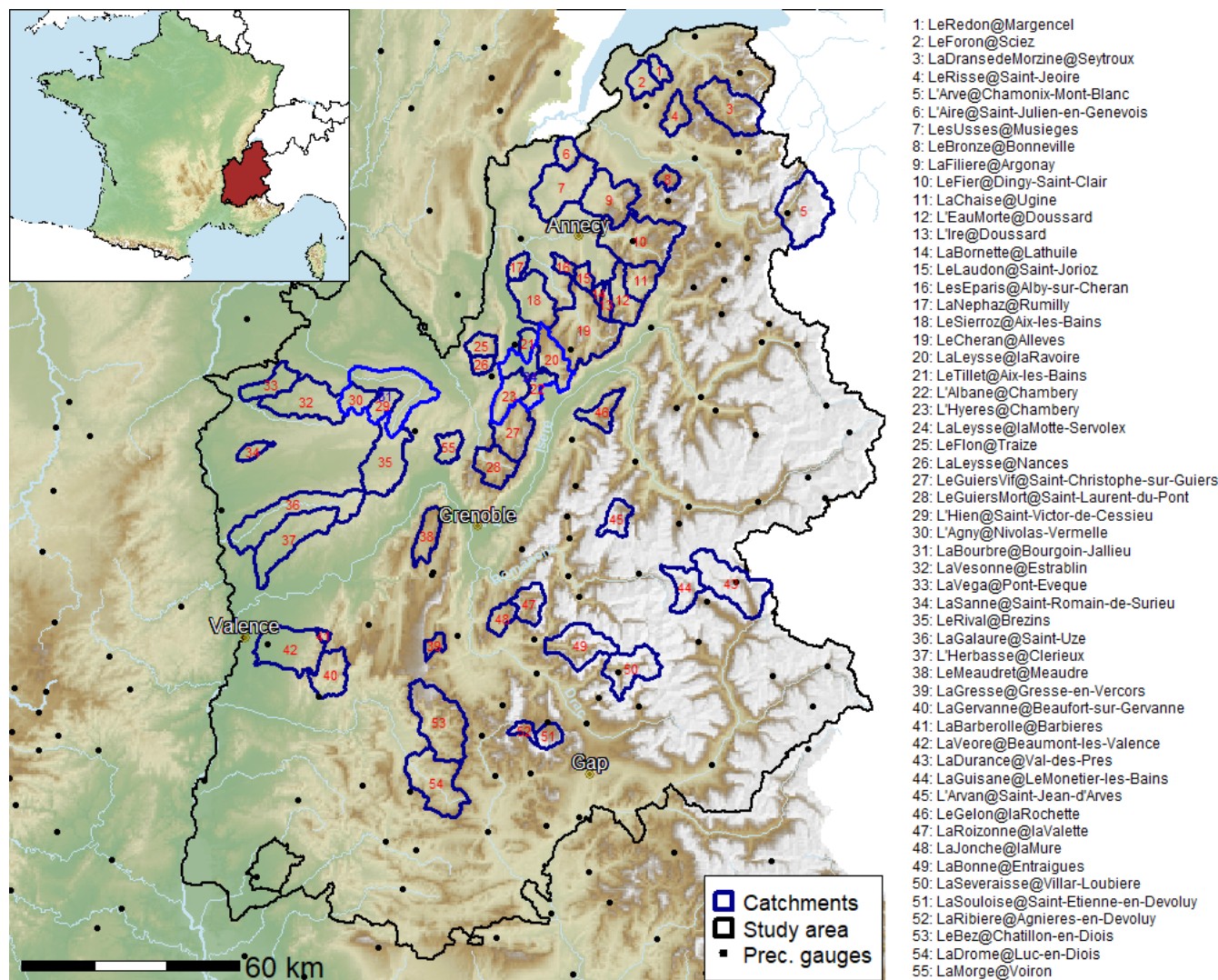

1: LeRedon@Margencel
2: LeForon@Sciez
3: LaDransedeMorzine@Seytroux
4: LeRisse@Saint-Jeoire
5: L'Arve@Chamonix-Mont-Blanc
6: L'Aire@Saint-Julien-en-Genevois
7: LesUsses@Musieges
8: LeBronze@Bonneville
9: LaFiliere@Argonay
10: LeFier@Dingy-Saint-Clair
11: LaChaise@Ugine
12: L'EauMorte@Doussard
13: L'Ire@Doussard
14: LaBornette@Lathuile
15: LeLaudon@Saint-Jorioz
16: LesEparis@Alby-sur-Cheran
17: LaNephaz@Rumilly
18: LeSierroz@Aix-les-Bains
19: LeCheran@Alleves
20: LaLeysse@laRavoire
21: LeTillet@Aix-les-Bains
22: L'Albane@Chambery
23: L'Hyeres@Chambery
24: LaLeysse@laMotte-Servolex
25: LeFlon@Traize
26: LaLeysse@Nances
27: LeGuiersVif@Saint-Christophe-sur-Guiers
28: LeGuiersMort@Saint-Laurent-du-Pont
29: L'Hien@Saint-Victor-de-Cessieu
30: L'Agny@Nivolas-Vermelle
31: LaBourbre@Bourgoin-Jallieu
32: LaVesonne@Estrablin
33: LaVega@Pont-Eveque
34: LaSanne@Saint-Romain-de-Surieu
35: LeRival@Brezins
36: LaGalaure@Saint-Uze
37: L'Herbasse@Clerieux
38: LeMeaudret@Meaudre
39: LaGresse@Gresse-en-Vercors
40: LaGervanne@Beaufort-sur-Gervanne
41: LaBarberolle@Barbieres
42: LaVeore@Beaumont-les-Valence
43: LaDurance@Val-des-Pres
44: LaGuisane@LeMonetier-les-Bains
45: L'Arvan@Saint-Jean-d'Arves
46: LeGelon@laRochette
47: LaRoizonne@laValette
48: LaJonche@laMure
49: LaBonne@Entraigues
50: LaSeveraisse@Villar-Loubiere
51: LaSouloise@Saint-Etienne-en-Devoluy
52: LaRibiere@Agnieres-en-Devoluy
53: LeBez@Chatillon-en-Dios
54: LaDrome@Luc-en-Diois
55: LaMorge@Voiron

**Figure 1.** Location of the 55 catchments (in dark blue) in the study area (in black) and of 149 hourly precipitation gauges available in this region (black dots).

**Table 1.** Characteristics of the 55 catchments: Number; Name (river and outlet); Area (km$^2$); Median elevation (m); Minimum elevation (m); Maximum elevation (m); Percentage of the catchment covered by ice; Percentage of the catchment characterized as karstic or supplying karstic sources.

| Number | Name | Area | Med. elev. | Min. elev. | Max. elev. | %ice | %karst |
|---|---|---|---|---|---|---|---|
| 1 | LeRedon@Margencel | 30 | 605 | 409 | 1519 | 0 | 21 |
| 2 | LeForon@Sciez | 59 | 569 | 387 | 1451 | 0 | 0 |
| 3 | LaDransedeMorzine@Seytroux | 172 | 1424 | 694 | 2449 | 0 | 5 |
| 4 | LeRisse@Saint-Jeoire | 55 | 1111 | 539 | 1939 | 0 | 82 |
| 5 | L'Arve@Chamonix-Mont-Blanc | 201 | 2510 | 1020 | 4187 | 28 | 0 |
| 6 | L'Aire@Saint-Julien-en-Genevois | 44 | 632 | 436 | 1380 | 0 | 18 |
| 7 | LesUsses@Musieges | 186 | 663 | 343 | 1379 | 0 | 16 |
| 8 | LeBronze@Bonneville | 29 | 1438 | 455 | 2335 | 0 | 100 |
| 9 | LaFiliere@Argonay | 155 | 858 | 488 | 1997 | 0 | 40 |
| 10 | LeFier@Dingy-Saint-Clair | 229 | 1220 | 514 | 2589 | 0 | 100 |
| 11 | LaChaise@Ugine | 78 | 982 | 429 | 2380 | 0 | 100 |
| 12 | L'EauMorte@Doussard | 92 | 1059 | 456 | 2297 | 0 | 100 |
| 13 | L'Ire@Doussard | 27 | 1241 | 469 | 2159 | 0 | 100 |
| 14 | LaBornette@Lathuile | 12 | 1108 | 467 | 1882 | 0 | 100 |
| 15 | LeLaudon@Saint-Jorioz | 29 | 928 | 468 | 1750 | 0 | 100 |
| 16 | LesEparis@Alby-sur-Cheran | 24 | 645 | 417 | 1693 | 0 | 31 |
| 17 | LaNephaz@Rumilly | 29 | 550 | 326 | 1008 | 0 | 19 |
| 18 | LeSierroz@Aix-les-Bains | 133 | 521 | 245 | 1556 | 0 | 23 |
| 19 | LeCheran@Alleves | 261 | 1149 | 577 | 2194 | 0 | 100 |
| 20 | LaLeysse@laRavoire | 95 | 1065 | 301 | 1828 | 0 | 100 |
| 21 | LeTillet@Aix-les-Bains | 33 | 352 | 251 | 1502 | 0 | 20 |
| 22 | L'Albane@Chambery | 47 | 466 | 279 | 1550 | 0 | 100 |
| 23 | L'Hyeres@Chambery | 84 | 683 | 264 | 1668 | 0 | 73 |
| 24 | LaLeysse@laMotte-Servolex | 289 | 684 | 239 | 1828 | 0 | 80 |
| 25 | LeFlon@Traize | 45 | 593 | 293 | 1477 | 0 | 29 |
| 26 | LaLeysse@Nances | 28 | 524 | 379 | 1373 | 0 | 19 |
| 27 | LeGuiersVif@Saint-Christophe-sur-Guiers | 120 | 1179 | 407 | 2043 | 0 | 100 |
| 28 | LeGuiersMort@Saint-Laurent-du-Pont | 92 | 1232 | 400 | 2068 | 0 | 98 |
| 29 | L'Hien@Saint-Victor-de-Cessieu | 50 | 518 | 352 | 688 | 0 | 0 |
| 30 | L'Agny@Nivolas-Vermelle | 58 | 498 | 292 | 684 | 0 | 0 |
| 31 | LaBourbre@Bourgoin-Jallieu | 303 | 455 | 246 | 753 | 0 | 0 |
| 32 | LaVesonne@Estrablin | 173 | 424 | 219 | 602 | 0 | 0 |
| 33 | LaVega@Pont-Eveque | 81 | 297 | 174 | 423 | 0 | 0 |
| 34 | LaSanne@Saint-Romain-de-Surieu | 30 | 380 | 241 | 463 | 0 | 0 |
| 35 | LeRival@Brezins | 175 | 490 | 368 | 784 | 0 | 0 |
| 36 | LaGalaure@Saint-Uze | 233 | 400 | 159 | 724 | 0 | 0 |
| 37 | L'Herbasse@Clerieux | 195 | 362 | 140 | 629 | 0 | 0 |
| 38 | LeMeaudret@Meaudre | 74 | 1249 | 958 | 1686 | 0 | 100 |
| 39 | LaGresse@Gresse-en-Vercors | 24 | 1401 | 1116 | 2325 | 0 | 100 |
| 40 | LaGervanne@Beaufort-sur-Gervanne | 104 | 835 | 314 | 1570 | 0 | 100 |
| 41 | LaBarberolle@Barbieres | 10 | 739 | 437 | 1286 | 0 | 4 |
| 42 | LaVeore@Beaumont-les-Valence | 190 | 271 | 126 | 1238 | 0 | 9 |
| 43 | LaDurance@Val-des-Pres | 193 | 2291 | 1361 | 3082 | 0 | 0 |
| 44 | LaGuisane@LeMonetier-les-Bains | 82 | 2399 | 1517 | 3601 | 2 | 0 |
| 45 | L'Arvan@Saint-Jean-d'Arves | 57 | 2027 | 1360 | 3440 | 4 | 0 |
| 46 | LeGelon@laRochette | 63 | 1013 | 330 | 2454 | 0 | 0 |
| 47 | LaRoizonne@laValette | 71 | 1736 | 935 | 2848 | 0 | 5 |
| 48 | LaJonche@laMure | 47 | 1038 | 876 | 2290 | 0 | 77 |
| 49 | LaBonne@Entraigues | 142 | 1952 | 778 | 3540 | 0 | 7 |
| 50 | LaSeveraisse@Villar-Loubiere | 130 | 2182 | 1020 | 3633 | 1 | 0 |
| 51 | LaSouloise@Saint-Etienne-en-Devoluy | 39 | 1770 | 1262 | 2620 | 0 | 0 |
| 52 | LaRibiere@Agnieres-en-Devoluy | 24 | 1683 | 1252 | 2587 | 0 | 1 |
| 53 | LeBez@Chatillon-en-Diois | 225 | 1234 | 559 | 2029 | 0 | 100 |
| 54 | LaDrome@Luc-en-Diois | 197 | 1000 | 540 | 1712 | 0 | 99 |
| 55 | LaMorge@Voiron | 46 | 532 | 268 | 942 | 0 | 14 |

## 2.2 Meteorological data

In this study, precipitation and temperature reanalysis available at a high spatial resolution and at an hourly time scale for the period 1997-2017 are selected in order to provide homogeneous meteorological forcings (i.e. without missing data) with an adequate representation of the meteorological dynamics at the catchment scale. Downscaling methods (Parkes et al., 2012; Breinl and Di Baldassarre, 2019) and conditional simulation (Bárdossy and Pegram, 2016) can be applied to obtain meteorological forcings at the appropriate resolution for the hydrological model. In this study, we rely on the spatial resolution provided by the reanalysis and apply disaggregation methods to obtain precipitation data at the hourly scale for the SPAZM reanalysis.

Table 2 presents the key features of the atmospheric reanalysis products considered in this study. They all cover the period 1997-2017. Four types of hourly precipitation data are used to evaluate the influence of the different precipitation forcings: ERA5-Land (Muñoz-Sabater et al., 2021), COMEPHORE (Champeaux et al., 2009), and two sets of SPAZM precipitation reanalysis (Gottardi et al., 2012) disaggregated from a daily to an hourly time step. As we focus on the influence of the different kinds of precipitation forcings, a unique source of temperature data is considered in all hydrological applications: the SPAZM temperature reanalysis disaggregated with hourly SAFRAN temperature.

**Table 2.** Key features of atmospheric reanalysis products. The products written in bold are used as inputs in the present study.

| Type of data | Name | Spatial and temporal resolution | Data assimilation | Post-treatment | Final resolution |
|---|---|---|---|---|---|
| Precipitation | **ERA5-Land** | 9 km / hourly | Satellite | none | 9 km / hourly |
| | **COMEPHORE** | 1 km / hourly | Radar / Hourly and daily gauges | none | 1 km / hourly |
| | SPAZM | 1 km / daily | Daily gauges | temporal disaggregation: **SPAZM-g** (with gauges) and **SPAZM-c** (with both COMEPHORE and gauges) | 1 km / hourly |
| Temperature | SAFRAN | 8 km / hourly | Hourly temperatures | none | 8 km / hourly |
| | SPAZM | 1 km / daily | Min. and max. daily temperatures | temporal disaggregation: **SPAZM_temp** with hourly SAFRAN temperature | 1 km / hourly |

### 2.2.1 ERA5-Land

ERA5-Land (Muñoz-Sabater et al., 2021) is a reanalysis dataset derived from atmospheric data of the ERA5 reanalysis which provides hourly estimates of a large number of atmospheric, land and oceanic climate variables, at a 30 km resolution (Hersbach et al., 2020). ERA5 and ERA5-Land are produced by the European Center for Medium Range Weather Forecast (ECMWF) and can be easily accessed through the Copernicus Climate Data Store (CDS). As such, they are widely used and can be considered as a reference dataset for the analysis of land surface variables, e.g. streamflow (Alfieri et al., 2020; Xu et al.,

2022; Probst and Mauser, 2022). The ERA5-Land dataset provides many atmospheric and surface variables, including hourly precipitation amounts and temperature at a 9 km resolution. Unlike the other reanalysis considered in this study, ERA5-Land mostly assimilates satellite data and does not consider gauged measurements. It also has the coarsest spatial resolution.

### 2.2.2 SAFRAN

SAFRAN (Systeme d'Analyse Fournissant des Renseignement Atmosphériques à la Neige) is a widely used precipitation and temperature reanalysis available for France at an 8 km spatial resolution grid but with an effective resolution of around 30 km (Durand et al., 1993; Vidal et al., 2010). SAFRAN assimilates daily precipitation data from many stations but not at an hourly scale. While SAFRAN provides different meteorological variables at an hourly scale (temperature, wind, rain and snow intensities), the subdaily dynamics are inferred from vertical profiles obtained from a numerical weather model. As shown in Quintana-Seguí et al. (2008), the temporal dynamics at a subdaily scale are poorly described for precipitation, with many 6-hour blocks of constant precipitation derived from the vertical profile of humidity. As a consequence, we decided not to use hourly SAFRAN precipitation data. In this study, we only use SAFRAN hourly temperature data to disaggregate daily temperature data from the SPAZM reanalysis (see Appendix A2). SAFRAN assimilates in situ temperature measurements at an hourly time scale in about 4000 stations in France. Quintana-Seguí et al. (2008) show that its subdaily structure correlates well to observed data and can be considered to be adequate for our applications.

### 2.2.3 SPAZM

SPAZM is a reanalysis produced by EdF (Électricité de France) specifically for hydrological evaluations in a large southeastern part of France, where most of the catchments with hydroelectricity stakes are located. SPAZM combines in-situ measurements and meteorological guesses conditioned by the topography, the season, and the weather type of the target day (Gottardi et al., 2012). SPAZM precipitation is available at a daily time scale and at a 1 km$^2$ spatial resolution. SPAZM temperature is available at the same temporal and spatial resolutions (Gottardi, 2009) and provides minimum and maximum values for each day of the reanalysis.

Because an hourly resolution is required to perform the hydrological evaluation of the small catchments considered in this study, postprocessing of SPAZM data is performed to disaggregate precipitation and temperature data from daily to hourly time scales. For the disaggregation of SPAZM precipitation data, COMEPHORE and precipitation gauges are used to provide the temporal structures (see, e.g., Parkes et al., 2012, for the application of a similar strategy). In the following sections, SPAZM-c will refer to SPAZM precipitation data disaggregated with COMEPHORE or with gauged data, and SPAZM-g will refer to SPAZM precipitation disaggregated with gauged data only (see Appendix A1 for further details). These gauged data are composed of 149 precipitation gauges belonging to EdF and Météo-France and which are located in the study area (see Fig. 1).

For the disaggregation of SPAZM temperature data, the hourly temporal structure from the SAFRAN reanalysis provides the subdaily dynamics, while the daily minimum and maximum temperatures are obtained from SPAZM (Gottardi, 2009). Further details are provided in Appendix A2. Note that, although ERA5-Land and SAFRAN also provide hourly temperature data, we choose to keep this SPAZM/SAFRAN composite reanalysis which has been shown to combine the strength of both

reanalyses on a mountainous catchment (Magand et al., 2018). The use of this single temperature forcing, combined with the four available precipitation forcings, also enables the assessment of the differences in the hydrological model performances related to the different precipitation inputs alone.

### 2.2.4 COMEPHORE

COMEPHORE (COmbinaison en vue de la Meilleure Estimation de la Precipitation HOraiRE) is a radar/gauge composite reanalysis of precipitation available at the spatial resolution of 1 km$^2$ and at an hourly time step on the French territory (Champeaux et al., 2009) for the period 1997-2017. COMEPHORE assimilates in-situ measurements from many precipitation gauges (more than 4000 at a daily time step, and more than 1000 at an hourly scale) and radar data. However, it must be noticed that the radar network significantly evolved during the period 1997-2017. In particular, the quality of radar measurements in the French Alps is rather poor before 2006 due to a lack of coverage by the French radar network (see Fig. 1 in Fumière et al., 2020). In addition, the methodology applied to merge radar and in-situ measurements is also different for the periods 1997-2006 and 2007-2017 with a different treatment of convective and stratiform precipitation events after 2007 (Laurantin, 2008).

For each of the 55 catchments and for each hour of the period 1997-2017, four different precipitation forcings are thus used as inputs for the two hydrological models considered in this study, corresponding to ERA5-Land, SPAZM-g, SPAZM-c and COMEPHORE, along with one temperature forcing corresponding to the SPAZM/SAFRAN composite reanalysis.

## 2.3 Hydrological models

### 2.3.1 MORDOR-SD

MORDOR hydrological model (Garçon, 1996) is the operational hydrological model of Edf for about 30 years and is applied in different contexts (real-time forecastings, flood frequency analysis and continuous monitoring of water resources). Garavaglia et al. (2017) describe the last version of the semi-distributed model MORDOR-SD with a spatialization of the main meteorological forcings and hydrological processes by elevation band. In particular, MORDOR-SD represents the accumulation and melting of ice and snow cover in each of these elevation bands.

MORDOR-SD considers mean areal precipitation (MAP) and temperature (MAT) values as inputs, which are obtained by averaging the precipitation or temperature of all pixels belonging to the catchment. If a pixel only partially covers a catchment, a weight corresponding to the surface covered by the catchment is assigned. The potential evapotranspiration PET (mm) is driven by MAT values and is obtained with the formula proposed by Oudin et al. (2005).

The parameterization of the version applied in this study contains 12 free parameters described in Table 3. In this study, four catchments with a significant fraction of their surface covered with ice (1%, 2%, 4% and 28%, respectively) require 2 additional parameters related to ice melting. The parameter $cp$ is a correction factor of the total amount of precipitation and directly affects the water balance. A constraint $zmax = umax$ for the maximum capacity of the capillarity storage $zmax$ and maximum capacity of the root zone $umax$ is applied, as these two parameters are strongly interrelated.

In this study, MORDOR-SD is calibrated using the procedure described by Paquet et al. (2013) and Garavaglia et al. (2017) which applies a genetic algorithm to find the parameters that maximize a multi-criterion objective function. This objective function minimizes the difference between observed and simulated streamflow time series, seasonal streamflows and flow

duration curves, these differences being quantified with the Kling-Gupta efficiency (KGE) criteria (Gupta et al., 2009).

**Table 3.** MORDOR-SD free parameters, units, range and description.

| Parameter | Units | Prior range | Description |
|---|---|---|---|
| cp | - | [0.1,3] | Precipitation correction factor |
| gtz | $^{\circ}$C 100 m$^{-1}$ | [-0.8,-0.4] | Air temperature gradient |
| umax | mm | [30,300] | Maximum capacity of the root zone |
| lmax | mm | [30,300] | Maximum capacity of the hillslope zone |
| cel | km h$^{-1}$ | [0.1,5] | Wave celerity |
| kdif | km | [0.1,5] | $dif/cel$ ratio where $dif$ is the wave diffusion in km$^2$h$^{-1}$ |
| evl | - | [1.5,4] | Outflow exponent of storage related to the hillslope zone |
| kr | - | [0.1,0.9] | Runoff coefficient |
| lkn | log(mm h$^{-1}$) | [-6,-1] | logarithm of the outflow coefficient of base flow storage |
| kf | mm $^{\circ}$C$^{-1}$ day$^{-1}$ | [0,5] | Constant part of melting coefficient |
| eft | $^{\circ}$C | [-3,3] | Additive correction of snowpack temperature |
| efp | $^{\circ}$C | [-3,3] | Additive correction of temperature for rain-snow partitioning |
| kg | mm $^{\circ}$C$^{-1}$ | [2,8] | Fixed part of the glacial melting coefficient |
| efg | $^{\circ}$C | [-3,3] | Additive correction of melting ice temperature |

### 2.3.2 SMASH

SMASH is a computational software framework dedicated to *Spatially distributed Modeling and data ASsimilation for Hydrology*. It provides a flexible hydrological spatially distributed modelling framework, capable of operating at high spatio-temporal resolution. It includes many functionalities for parameter sensitivity analysis, uniform, and spatially distributed parameter

calibration methods as well as variational data assimilation algorithms (Jay-Allemand et al., 2020).

SMASH is the result of work carried out at INRAE of Aix-en-Provence in the fields of flood forecasting (Javelle et al., 2016) and low water levels modeling in France (Folton and Arnaud, 2020). Based on a conceptual representation of the dominant hydrological processes, SMASH is a continuous distributed model that enables to represent, on each grid cell, different hydrological functions at the user's choice: snow accumulation and melting, production, transfer within the grid cell, and runoff

routing between grid cells. It uses spatially distributed meteorological forcings and hydrometric observations.

The model has been developed with the objective to maintain a relative parametric parsimony, in order to facilitate its regionalization and allow its application to ungauged catchments. The SMASH grid-based model structure implemented in this study combines the following components: i) the CemaNeige snow store introduced by Valéry (2010), ii) the production store and the transfer store of the GR4J model described in Perrin et al. (2003), iii) a second transfer store coupled with a direct

runoff branch, iv) the water exchange function described in Perrin et al. (2003) which enables to simulate losses or gains of water, which can be required in cases of non-conservative catchments (groundwater exchange) and/or data uncertainties, v) a simple cell to cell routing scheme (linear reservoir) to convey the discharge downstream following a drainage plan derived

from terrain elevation data. This model structure contains 8 free parameters described in Table 4 along with their calibration ranges. Further details about the model structure are provided in Appendix B.

The calibration algorithm used here to calibrate the SMASH model parameters is the simple steepest descent global-minimization algorithm. The objective function is the Kling-Gupta efficiency (KGE) criteria (Gupta et al., 2009) computed between the observed and simulated streamflow time series. In this study, SMASH is run at the spatial resolution of 1 km$^2$ and temporal resolution of 1 h, and for each catchment eight spatially uniform parameters are calibrated.

**Table 4.** SMASH free parameters, units, range and description.

| Parameter | Units | Prior range | Description |
|---|---|---|---|
| $c_p$ | mm | [1, 100000] | Capacity of the production reservoir |
| $c_{ft}$ | mm | [1, 1000] | Capacity of the $1^{st}$ transfer reservoir |
| $c_{st}$ | mm | [1, 10000] | Capacity of the $2^{nd}$ transfer reservoir |
| $l_r$ | min | [0.0001, 1000] | Linear routing reservoir parameter |
| $\alpha$ | - | [0, 1] | Partition parameter between the two transfer reservoirs |
| $tc$ | - | [0, 1] | Weighting coefficient for the thermal state of the snowpack |
| $mc$ | mm $^\circ$C$^{-1}$ hour$^{-1}$ | [0, 4] | Melting coefficient |
| $exc$ | mm hour$^{-1}$ | [-20, 20] | Water exchange parameter applied to the direct runoff branch and to the $1^{st}$ transfer reservoir |

## 3 Comparison of mean areal precipitation values

Figure 2 presents different statistics of mean areal precipitation obtained with the different precipitation products for the 55 catchments. The proportion of dry hours (Fig. 2a) varies between 0.93 and 0.97 and is roughly similar for the four different meteorological products and the different catchments. At a daily scale (Fig. 2b), ERA5-Land produces many more wet days than the other products. The average annual amounts cover the same range for SPAZM and ERA5-Land (Fig. 2c), which means that the annual amounts for ERA5-Land are obtained with a higher number of wet days with more moderate intensities (Fig. 2d). Annual amounts from COMEPHORE are about 14% smaller than SPAZM and ERA5-Land. The four products are equivalent concerning the number of days with more than 10 mm (Fig. 2e) but exhibit major discrepancies for the number of days with heavy precipitation (i.e. more than 50 mm, Fig. 2f), with 50% of the catchments with 1 to 2 days in average with SPAZM, and more than 5 in average for four catchments, while there is around 1 day with heavy precipitation with COMEPHORE, and around 0.25 with ERA5-Land. This agrees with the statistics of extreme mean areal precipitation values presented in Fig. 2g-i where ERA5-Land leads to much more moderate annual maxima than SPAZM and COMEPHORE at hourly and daily scales.

Obviously, the two versions of SPAZM lead to very similar results at a daily scale since the disaggregation method mostly impacts the subdaily statistics. A slight difference can be noticed for the maximum records of daily precipitation (Fig. 2i) due to the fact that SPAZM precipitation are recorded and disaggregated for days starting and ending at 6:00 in the morning while the statistics are computed on standard calendar days. At an hourly scale (Fig. 2g), the average annual maxima are higher when SPAZM is disaggregated with gauges only (SPAZM-g) than when it is disaggregated with COMEPHORE in priority. It can be

explained by the fact that the temporal structures from one gauge impact more than one pixel in SPAZM-g, so that many pixels of a catchment share exactly the same subdaily distribution, including the timing of the peak value.

These statistics are in line with previous evaluations of the different precipitation reanalysis which have concluded to the following deficiencies:

- **ERA5-Land:** ERA5-Land usually underestimates the largest intensities at hourly and daily scales. While ERA5-Land represents a good reference for mean statistics (annual amounts, seasonality), it seems unable to produce very intense precipitation events, probably due to the direct parameterisation of convection (Reder et al., 2022) and the absence of assimilation of ground measurements. The overestimation of the number of wet days was also denoted by Bandhauer et al. (2022).

- **SPAZM:** SPAZM has been shown to produce reliable amounts of precipitation at an annual scale and the pixel scale, by comparison to observed precipitation values (Penot, 2014). Hydrological evaluations have also shown that annual amounts of areal precipitation values are correctly estimated for large catchments (> 700 km$^2$). However, for smaller spatial scales, the variability of daily intensities and annual maxima tends to be underestimated (Penot, 2014).

- **COMEPHORE:** COMEPHORE tends to underestimate the daily precipitation amounts in mountainous areas (Roger, 2017) especially before 2006 and above 1000 m. As COMEPHORE does not integrate any additional constraint about the effect of the relief, the vertical profiles of annual precipitation amounts are almost flat. Roger (2017) also provides an evaluation of COMEPHORE in terms of return levels (10-year and 50-year) by comparison with return levels obtained from weather station measurements and more advanced techniques (Arnaud et al., 2008) and does not conclude to a particular under or over-estimation of these return levels. However, COMEPHORE has some limitations related to the assimilation of radar data, such as the radar signal attenuation by precipitation or beam blockage, which cannot be avoided in mountainous regions (Villarini and Krajewski, 2010).

## 4 Hydrological evaluation of meteorological forcings

### 4.1 Evaluation criteria

Many evaluation criteria have been proposed to assess the performances of hydrological models. Several studies have discussed the pros and cons of two popular criteria: The Nash-Sutcliffe efficiency (NSE Nash and Sutcliffe, 1970) and the Kling-Gupta Efficiency (KGE Gupta et al., 2009). Recently, Clark et al. (2021) emphasize the fact that these criteria rely on squared errors between simulated and observed streamflows and are subject to considerable sampling uncertainties. Large differences between observations and simulations are amplified by these squared errors. In this study, different performances are quantified with a modified NSE criterion ($mNSE$, see Krause et al., 2005) defined as follows:

$$mNSE(x) = 1 - \frac{\sum_t |x_{s,t} - x_{o,t}|}{\sum_t |x_{o,t} - \bar{x}_o|}, \tag{1}$$

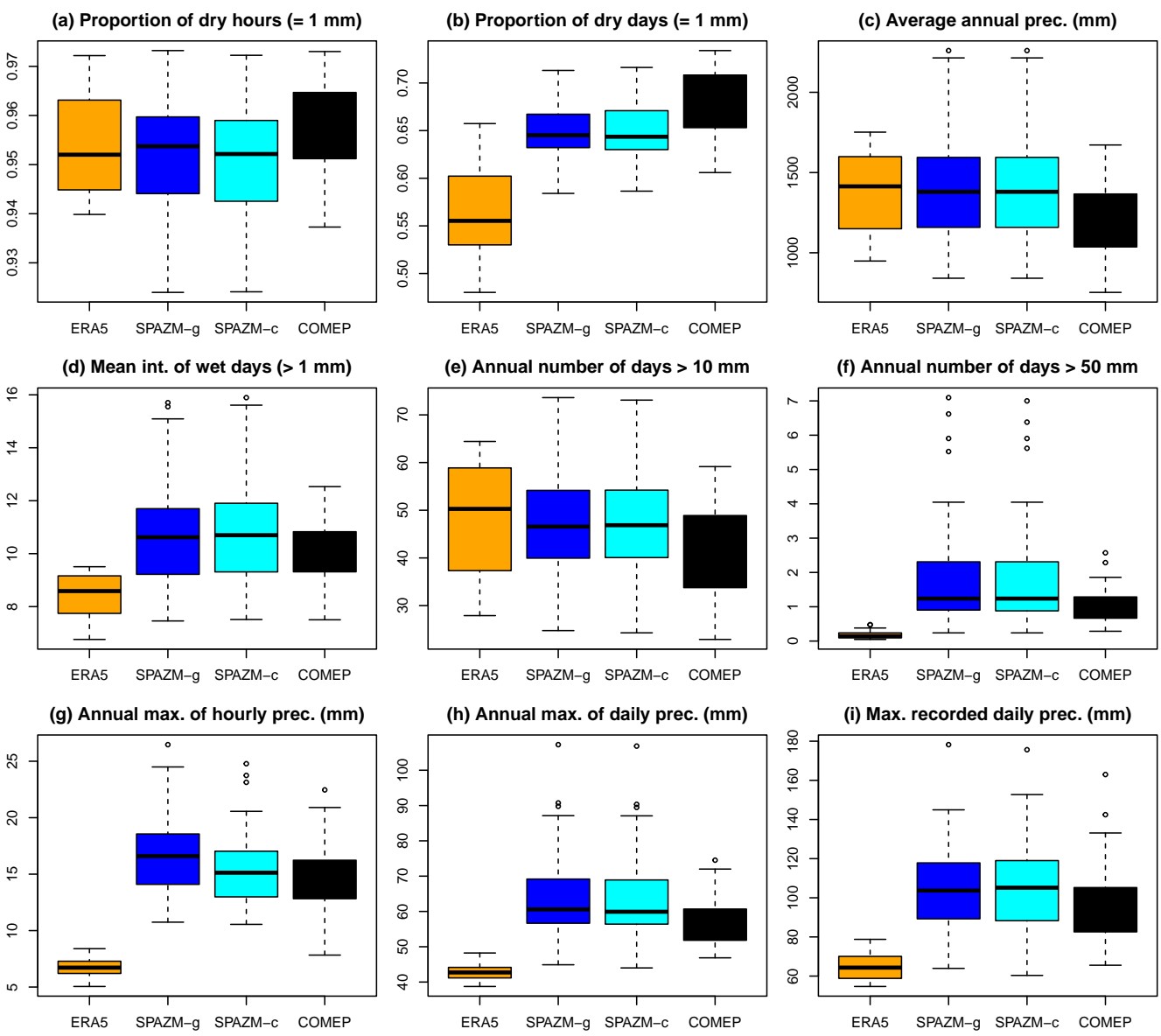

**Figure 2.** Boxplots of mean areal precipitation statistics for the different meteorological products and the 55 catchments. (a) Proportion of dry hours. (b) Proportion of dry days. (c) Average annual precipitation. (d) Mean intensity of wet days (i.e. days with more than 1 mm). (e) Average annual number of days with more than 10 mm. (f) Average annual number of days with more than 50 mm. (g) Mean annual maximum of hourly precipitation. (h) Mean annual maximum of daily precipitation. (i) Maximum record of daily precipitation for the period 1997-2017.

where $x$ is an hydrological signature which is either observed $(x_{o,t})$ at time $t$ or simulated $(x_{s,t})$. $\bar{x}_o$ is the average observed signature over the period. In what follows, the $mNSE$ criterion is applied to different types of signatures, e.g.:

- for the whole time series of flow (denoted by $Q$),

- for the inter-annual averages of hourly streamflow in order to evaluate the reproduction of the seasonal variation of observed streamflow (denoted by $Q_{sea}$).

This criterion can also be obtained for the different seasons. Here, we consider a first period from December to May in order to assess the reproduction of streamflows influenced by snow, and the rest of the year, from June to November.

An additional criterion, the Quantile Relative Error (QRE), is proposed to evaluate the reproduction of observed streamflows in terms of distribution: The QRE assesses the differences between observed and simulated quantiles and is defined as:

$$QRE = 1 - |\hat{F}_o^{-1}(p) - \hat{F}_s^{-1}(p)|/\hat{F}_o^{-1}(p), \tag{2}$$

where $\hat{F}_o$ and $\hat{F}_s$ are the empirical distributions of observed and simulated streamflows, respectively.

Finally, different criteria are proposed to assess the performances for the reproduction of flooding events (Lobligeois et al., 2014). Different methods have been proposed to extract floods from streamflow series, by identifying the peak flows, and the period of rising and declining limbs. These methods usually assume a threshold streamflow value. In our experience, the choice of this threshold is critical and can lead to very long flood events for slow-declining limbs. In this paper, we focus on the reproduction of the observed streamflows around the peak flow and extract 48-hour events centered on the 10 largest peak flows, with at least one week between each flood. Four criteria are considered:

- the $mNSE$ criteria is applied between the observed flood and the simulated streamflow of the corresponding period.

- the peak flows error ($PFE$) is defined as the relative error between observed and simulated peak flows, i.e. $PFE = |\max(q_{o,\mathbf{f}}) - \max(q_{s,\mathbf{f}})|/\max(q_{o,\mathbf{f}})$ where $\mathbf{f}$ is the flood period.

- the time to peak error ($TPE$) is the absolute difference between the time of observed and simulated peak flows in hours, i.e. $TPE = |argmax(q_{o,\mathbf{f}}) - argmax(q_{s,\mathbf{f}})|$, where $argmax$ indicates the time at which the observed and simulated flows reach their peak.

- the volume error ($VE$) is the sum of absolute differences between observed and simulated flows of the flood period, i.e. $VE = \sum(|q_{o,\mathbf{f}} - q_{s,\mathbf{f}}|/q_{o,\mathbf{f}})$.

The $PFE, TPE$ and $VE$ criteria are obtained for each selected flood. Their averages over the 10 selected floods are denoted respectively by $\overline{PFE}, \overline{TPE}$, and $\overline{VE}$. The $mNSE$ and $QRE$ criteria are positively oriented and have their maximum value equal to 1. $\overline{PFE}, \overline{TPE}$ and $\overline{VE}$ are negatively oriented and have an optimum value of zero.

## 4.2 Split-sample evaluation

A first split-sample evaluation (Klemeš, 1986) is performed by dividing the period 1997-2017 into two subperiods of equal lengths depending on the availability of the observed streamflows. 45 out of the 55 catchments cover the entire period 1997-2017 and the two hydrological models are calibrated on two periods of 9 years (1998-2007 and 2008-2017), one year being used as a warm-up period. The other 10 catchments cover at least the period 2004-2017. Model parameters are calibrated on the first half and criteria are computed on the second half, and conversely, for each type of meteorological forcings.

Figure 3 shows the different evaluation criteria obtained with the split-sample experiment on the validation periods, for the two hydrological models and the four types of precipitation forcings. The $mNSE$ applied to the entire series of streamflow (Fig. 3a-b) ranges between 0.2 and 0.55 for ERA5-Land, and 0.4 and 0.7 for COMEPHORE. A clear increase of the performances is obtained for SPAZM and COMEPHORE in summer and autumn (JJASON), compared to ERA5-Land. Concerning the seasonal flows (Fig. 3c-d), the differences are not as marked, the median criteria for the 55 catchments being around 0.5 for ERA5-Land, 0.6 for SPAZM and 0.6 for COMEPHORE in winter and spring (DJFMAM) and slightly higher in summer and autumn (JJASON). For these criteria, SMASH leads to large variations of performances across the catchments compared to MORDOR-SD. As discussed in Section 5.6 and shown in Figure S38 in the Supplement, interannual streamflows are underestimated by SMASH in winter and spring for many catchments and overestimated in summer and autumn.

The reproduction of the probability distribution of streamflow is evaluated in Fig. 3e-h for different levels of probability. The 0.50 and 0.99-quantiles (Fig. 3e-f) are adequately reproduced by all configurations, the median of the relative errors being less than 5% with MORDOR-SD and less than 10% for SMASH for most of the cases. For return periods of 2 and 10 years (Fig. 3g-h), the performances are variable according to the catchments, the 90% intervals of the $QRE$ indicating relative differences (1-$QRE$) varying between 60% and less than 5%. For these two return periods, median $QRE$ values with ERA5-Land are lower by 0.10-0.15 than with the other precipitation products.

The $mNSE$ applied to the 48-hour floods (Fig. 3i) shows a wide range of performances across the 55 catchments, the differences between observed and simulated streamflows being important for a large part of catchments. Indeed, the median $mNSE$ for floods is close to zero for ERA5-Land, and the 0.1-quantile is below zero in all cases, which means that the simulated streamflow is further away from the observed streamflow than the average observed streamflow during these periods. The peak flows errors ($\overline{PFE}$) are shown in Fig. 3j and range between 70% and 30% for ERA5-Land, 60% and 20% for SPAZM, and 50% and 20% for COMEPHORE. The differences in the timing of these peak flows (Fig. 3k) are of the order of a few hours, the median difference ($\overline{TPE}$) being around 4.5 hours for ERA5-Land and less than 3 hours for the other precipitation products.

Overall, we can see a clear hierarchy between the performances obtained with the four meteorological forcings, with increasing performances between ERA5-Land, SPAZM-g, SPAZM-c, and COMEPHORE. ERA5-Land has lower performances for all the considered criteria, the differences being less pronounced in terms of distribution (e.g. reproduction of the 0.50- and 0.99-quantiles). For the largest streamflow values, streamflows using ERA5-Land are not as close to the observed values as with the other products. A slight difference of performances can be seen between SPAZM-g and SPAZM-c, i.e. when SPAZM

is disaggregated with either gauged values or with radar and gauges information. The timing of the simulated flood peaks is better with SPAZM-c for some catchments (lower bound of the bar going from -6 hours to -5 hours). The gain provided by COMEPHORE compared to SPAZM is visible in terms of peak flow error (Fig. 3j) or volume error (Fig. 3l). While some differences of performances are observed between MORDOR-SD and SMASH for some criteria and some catchments (mainly in terms of seasonal streamflow, see Fig. 3c-d), it is interesting to note that the differences related to the choice of the precipitation forcings have very similar patterns for both hydrological models.

Figures 4 and 5 present the spatial variation of the $mNSE(Q)$ in winter and spring (DJFMAM) and of the $mNSE$ for the 10 largest floods, for the two hydrological models and the four types of precipitation forcings. Concerning the overall reproduction of streamflows in winter and spring, a clear west-east gradient is observed in all cases, lowest $mNSE$ values being reached in the plains at the Northwest of Grenoble, especially with ERA5-Land (similar results are obtained in summer and autumn, see Fig. S35 in the Supplement). The $mNSE$ for the 10 largest floods shown in Fig. 5 does not show the same patterns. A few catchments exhibit lower performances whatever the configuration. In particular, three catchments between Briançon and Grenoble lead to $mNSE$ values of -2.1,-3.8, and -1 with COMEPHORE and MORDOR-SD, and even lower values for the other cases.

Figure 6 illustrates a meteorological event on September 6th, 2008 which led to major flood impacts (cars swept away, roof collapses) in the west of the domain considered in this study (https://youtu.be/Ngk2eV_WJk8). In particular, in the small village of Saint-Donat-sur-l'Herbasse, a flood wave of 1m70 crossed the main street and damaged cars and houses. Cumulative amounts of precipitation over our study area reached 140 mm, with maximum hourly intensities around 20 mm on the morning of September 6th, 2008. For the catchment LaGalaure@Saint-Uze, all simulated streamflows underestimate the observed peak flow at 250 m³/s, but this underestimation is less severe when COMEPHORE is used as input of the hydrological models, followed by SPAZM-c. COMEPHORE leads to a reasonable reproduction of observed peak flows for the other catchments. In this example, we can notice the different results obtained with SPAZM-c and SPAZM-g for two catchments: LaGalaure@Saint-Uze and L'Herbasse@Clerieux, where SPAZM-c, in coherence with COMEPHORE, provides a larger hourly intensity of precipitation two hours before SPAZM-g and simulates highest peak flows. In this case, the timing of the precipitation event provided by the radar information seems to produce a better reproduction of the observed flood.

Figure 7 illustrates the flooding event of October 22-23, 2013 consecutive to a series of several storm cells organized along a southwest-northeast axis crossing the Départements of Ardèche and Drôme (https://youtu.be/sId0smhhF70). Cumulative amounts exceeded 200 mm in the southwest part of our study area according to COMEPHORE. At the time of this event, the soil was probably saturated by a recent precipitation event on 19-20 October 2013 (50-100 mm). For two catchments that have reached the highest observed peak flows for this event (L'Herbasse@Clerieux and LaGalaure@Sainte-Uze), COMEPHORE provides much more precipitation than ERA5-Land and SPAZM and the simulated streamflow obtained with this reanalysis match almost perfectly the observed streamflow. SPAZM leads to a very large underestimation of the peak flow. ERA5-Land, in this case, provides little precipitation in this area. This event exemplifies the added value of radar information.

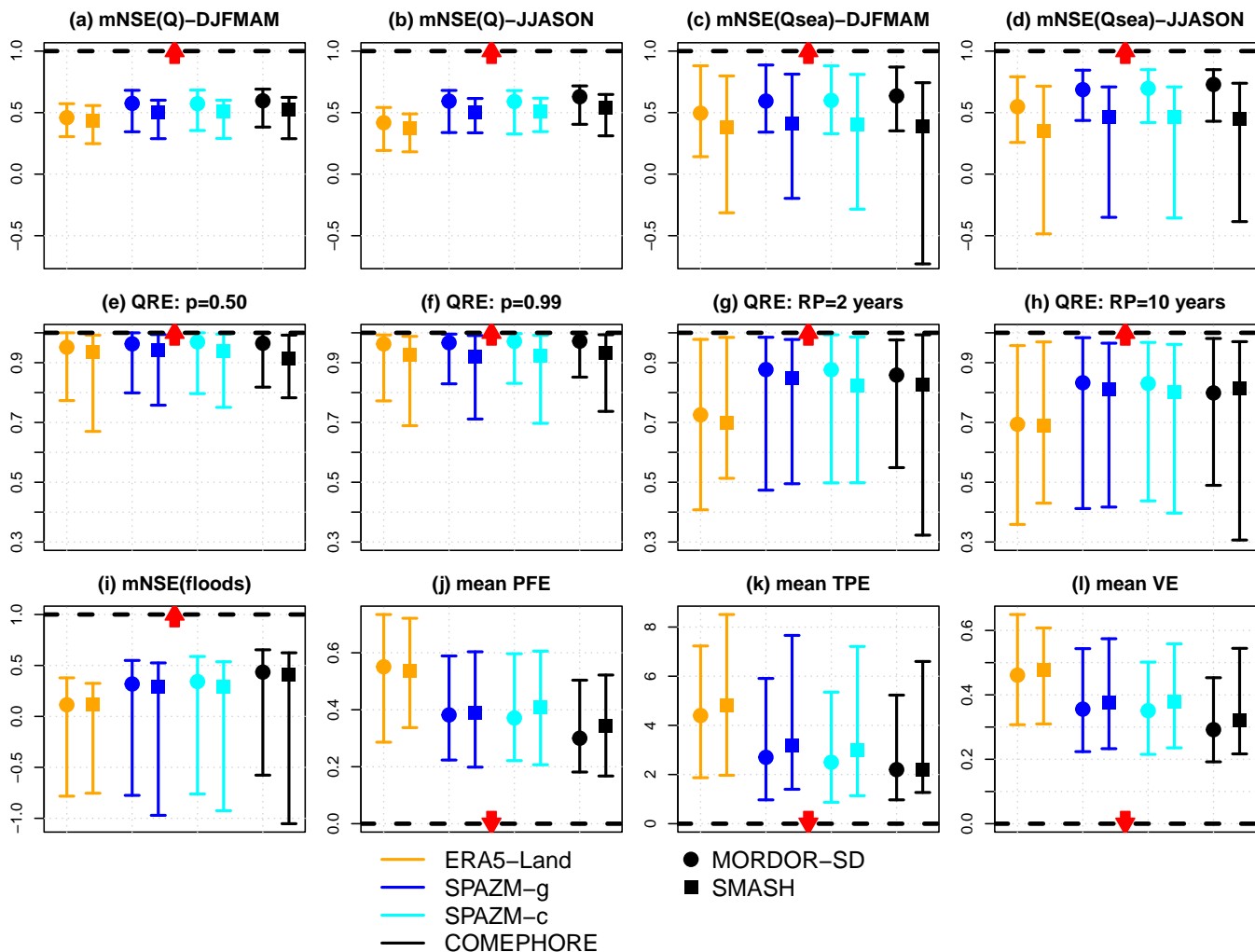

**Figure 3.** Evaluation criteria for the two hydrological models and the four types of precipitation forcings, computed on the two validation periods of the split-sample experiment. The colored error bars indicated the 0.10 and 0.90 quantiles over the 55 catchments, and the symbols indicate the median values. (a-b) $mNSE$ criteria for the streamflows $Q$ in DJFMAM and JJASON. (c-d) $mNSE$ criteria for the seasonal streamflow $Q_{sea}$ in DJFMAM and JJASON. (e-h) $QRE$ criteria for the median streamflows ($p = 0.50$), 0.99-quantiles, and streamflows associated with return periods of 2 and 10 years. (i-l) $mNSE$ for the 10 largest floods, peak flows error ($\overline{PFE}$), time to peak error ($\overline{TPE}$) and volume error ($\overline{VE}$). Optimum values are indicated with horizontal black dashed lines and red arrows indicate if the criteria are negatively or positively oriented.

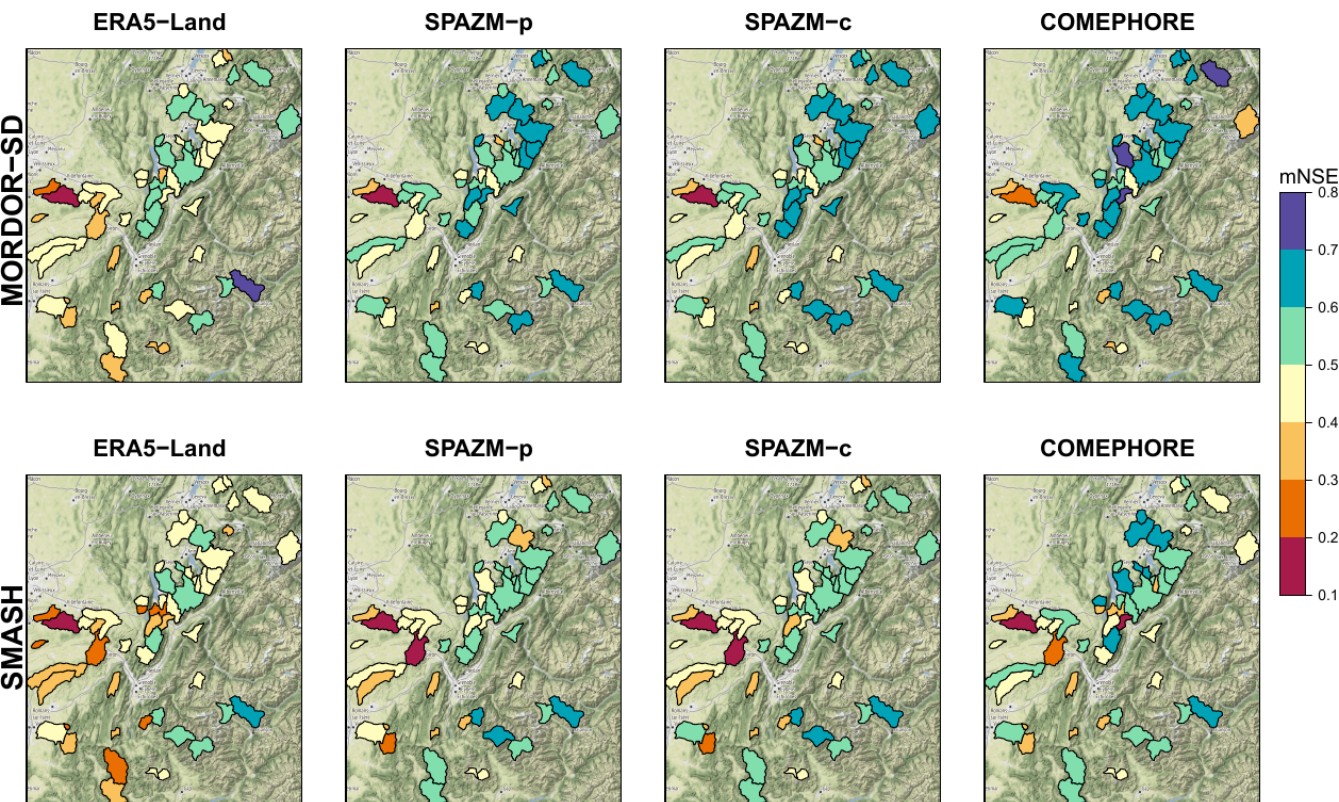

**Figure 4.** Maps of the $mNSE(Q)$ in winter and spring (DJFMAM) obtained with MORDOR-SD (top row) and SMASH (bottom row) and the four types of precipitation forcings. Map tiles by Stamen Design, under CC BY 3.0. Data by OpenStreetMap, under ODbL.

## 5  Discussion

### 5.1  Reproduction of the water balance

A key property of precipitation inputs is their ability to balance the amount of water measured by the streamflow at aggregated

time scales (annual or monthly scales). In order to correct the water balance, SMASH and MORDOR-SD include correction parameters, $exc$ and $cp$, respectively. the SMASH parameter $exc$ enables water exchanges but is only applied to a part of the streamflow production (direct runoff branch). For MORDOR-SD, the parameter $cp$ is a multiplication factor that is applied to the precipitation inputs and can be interpreted in terms of correction of the water balance. A $cp$ value below 1 means that total precipitation amounts do not lead to a correct water balance, and must be reduced. It can be due to various reasons.

A possible explanation is that water is exiting the basin as groundwater. This seems to be the case for some catchments of the study areas located in the plains around Valence and the northwest of Grenoble, which have important exchanges with groundwater. Two of these catchments located in the "4 vallées" area have important infiltration of surface water upstream and an important contribution of groundwater to the streamflow downstream. For these catchments, the runoff ratio (i.e. the

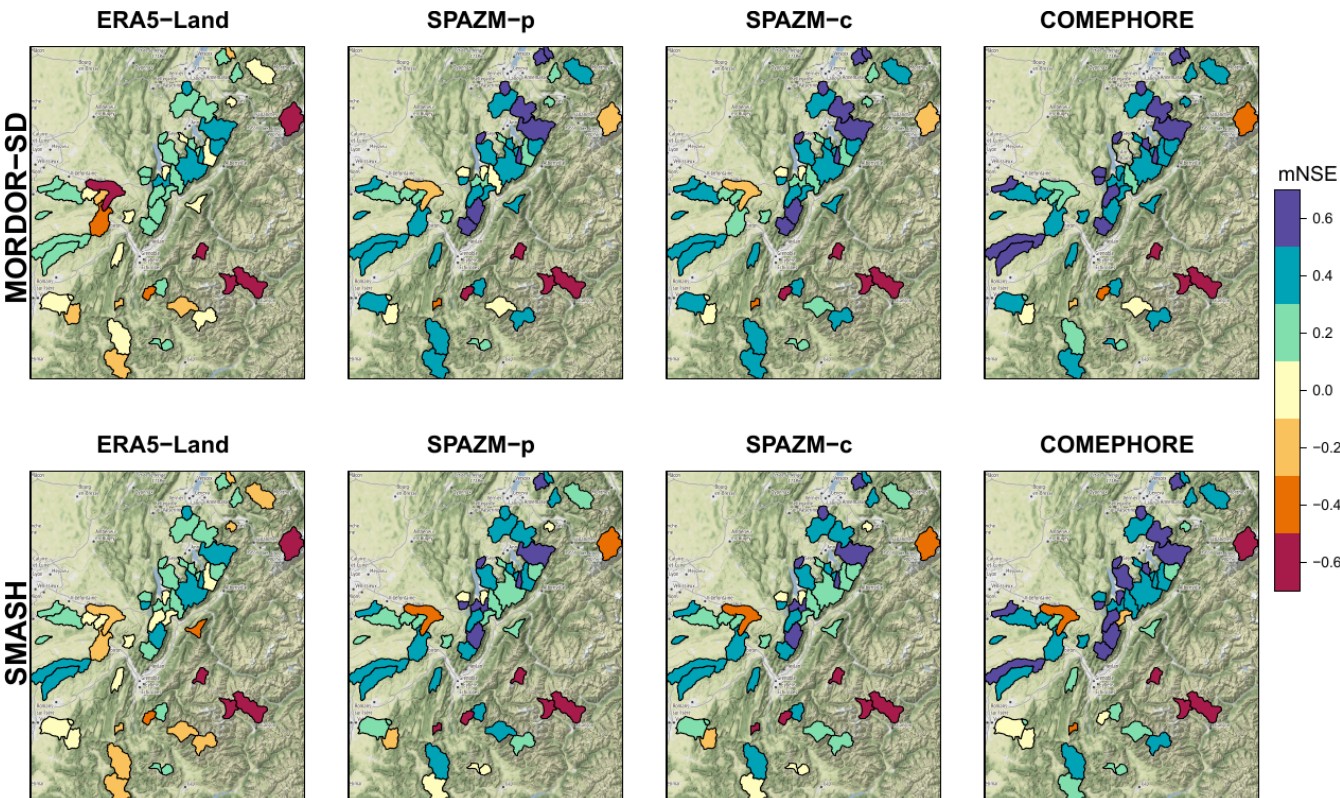

**Figure 5.** Maps of the $mNSE$ for the 10 largest floods obtained with MORDOR-SD (top row) and SMASH (bottom row) and the four types of precipitation forcings. A few values exceed the lower bound of -0.7 (dark brown). Map tiles by Stamen Design, under CC BY 3.0. Data by OpenStreetMap, under ODbL.

proportion of precipitation that does not infiltrate and is not taken up by evapotranspiration, and thus ends up as runoff) can
vary by a factor of 5 (Brenot and Dupré la Tour, 2010). An additional factor is the high presence of karstic areas. Figure
S1 in the supplementary materials shows the percentage of areas characterized as karstic or supplying karstic sources, for
each catchment of this study (Brugeron et al., 2018). This percentage exceeds 0.8 for many catchments located along the
Drac and Isère rivers in the Vercors, Chartreuse, and Bauges massifs. On the contrary, when a $cp$ value exceeds 1, it means
that insufficient precipitation amounts are provided, which is rarely explained by the particular geology of the catchment. A
probable explanation is an underestimation of accumulated precipitation amounts in the corresponding reanalysis.

Figure 8a shows the map of the ratios between the mean annual precipitation values from SPAZM and COMEPHORE, for
each pixel of the domain. These ratios are greater than 1 at high elevations and less than 1 in the valleys. For the catchments
located at high elevations, SPAZM clearly leads to larger accumulated amounts than COMEPHORE. Figure 8b shows the
values of the $cp$ parameter as a function of the median elevation of each catchment, for the four different precipitation reanalysis.
The catchments with a high percentage of karst (greater than 20%) are masked. $cp$ values range between 0.6 and 1 for median

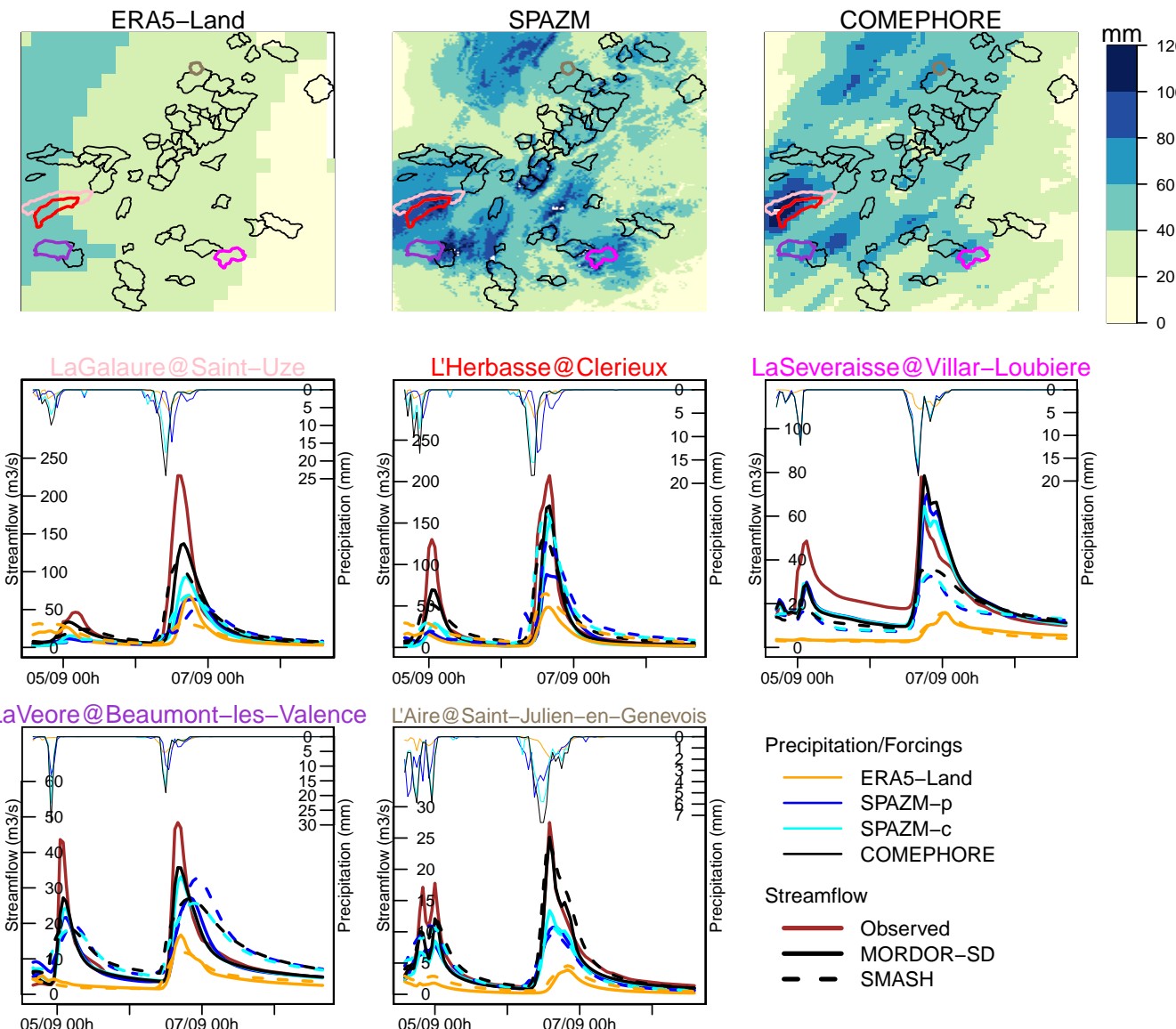

**Figure 6.** Cumulative amount of precipitation between 05/09/2008 06:00 and 07/09/2008 06:00 for ERA5-Land, SPAZM and COMEPHORE (top row) and time series of precipitation (thin lines) and streamflow (thick lines) for five catchments which have reached the highest observed peak flows for this event (middle and bottom rows).

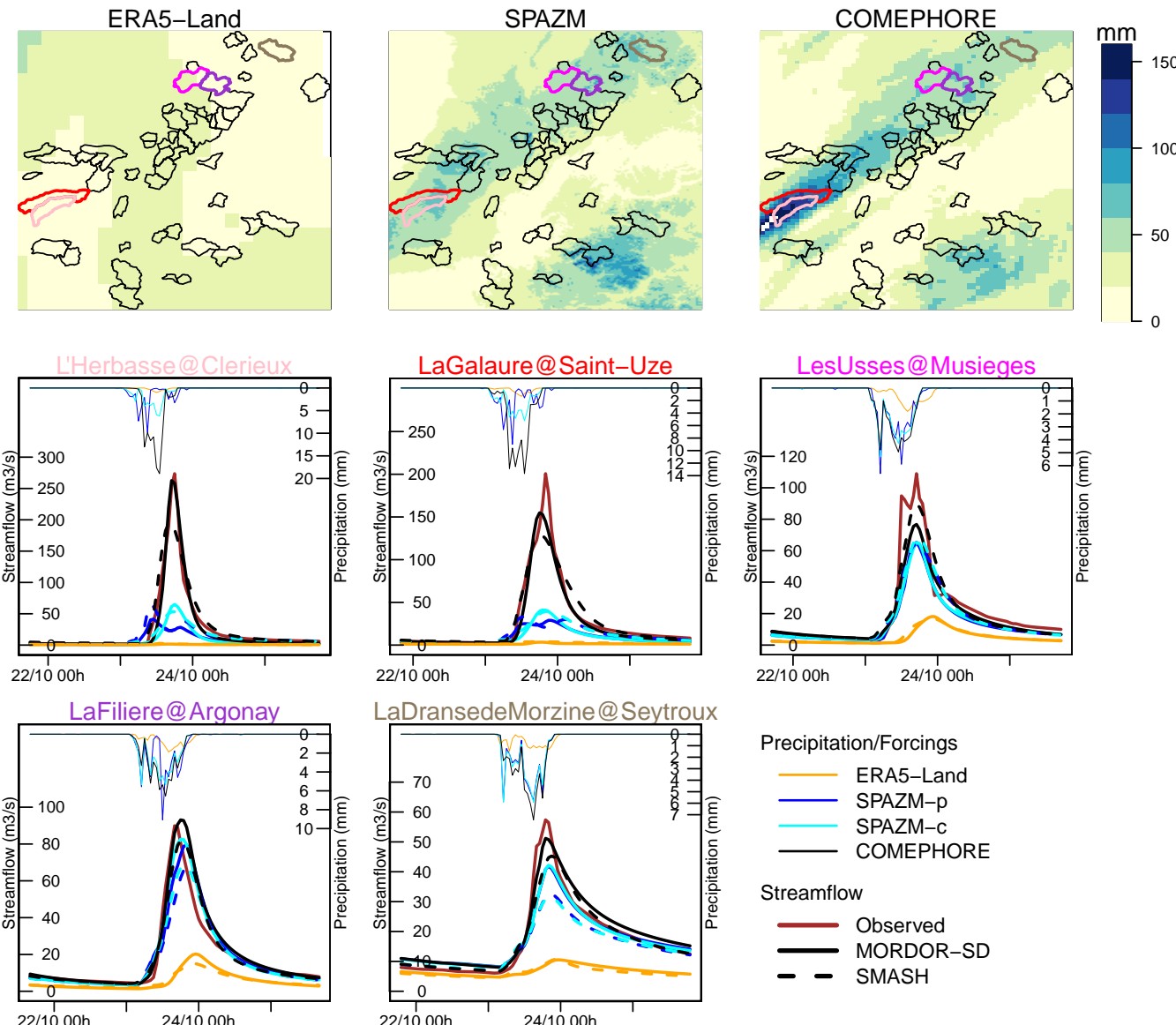

**Figure 7.** Cumulative amount of precipitation between 22/10/2013 06:00 and 24/10/2013 06:00 for ERA5-Land, SPAZM and COMEPHORE (top row) and time series of precipitation (thin lines) and streamflow (thick lines) for five catchments which have reached the highest observed peak flows for this event (middle and bottom rows).

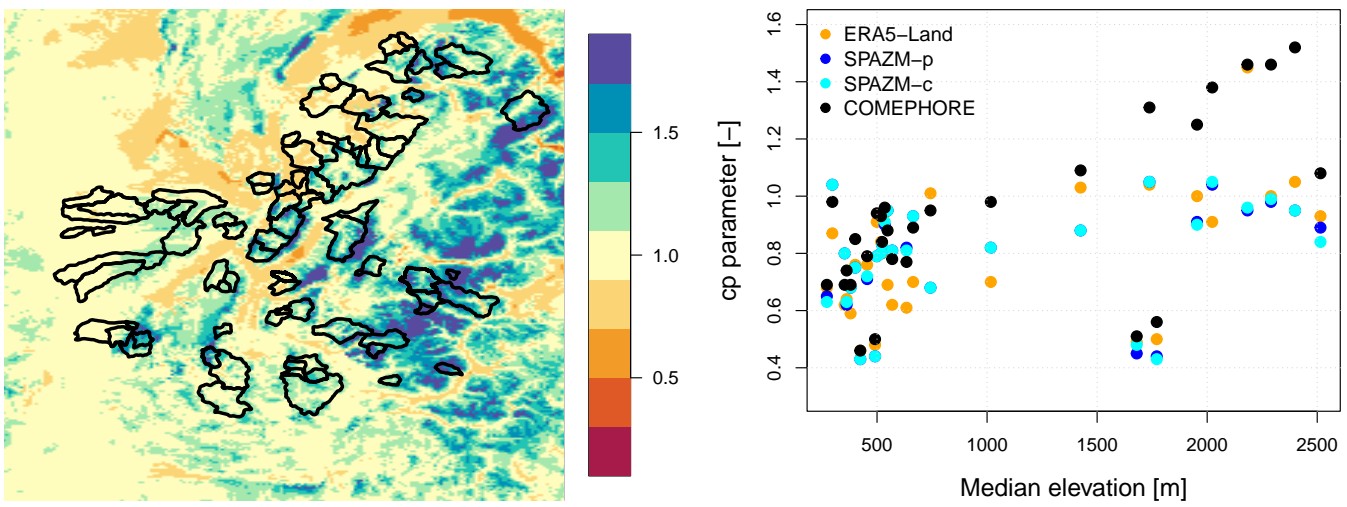

**(a) Ratio SPAZM/COMEPHORE of mean annual Prec.**

**(b) cp parameter versus elevation**

**Figure 8.** (a) Ratio of mean annual precipitation between SPAZM and COMEPHORE. The catchment boundaries are indicated in black. (b) Values of the $cp$ parameter as a function of the median elevation of each catchment with a percentage of karst below 20%, for the four different precipitation reanalysis.

elevations below 1000 m, probably confirming large contributions of precipitation to groundwater for some catchments located in the plains. Above 1000 m, it is interesting to note that $cp$ is close to 1 when SPAZM and ERA5-Land are used as inputs. COMEPHORE seems to underestimate precipitation amounts in high-elevation catchments ($cp > 1$), confirming the findings of Roger (2017) detailed in Section 3.

## 5.2 Atmospheric forcings obtained from numerical weather models and satellite data

ERA5-Land, as most reanalysis at the planetary scale, mainly results from a numerical model representing the atmospheric circulation, constrained by satellite observations. As such, it represents the main dynamics of the atmosphere and provides a reasonable representation of the main hydrometeorological fluxes. For example, Figs. 2c and 8b show that annual precipitation amounts obtained with SPAZM and ERA5-Land are similar and do not need to be corrected at high elevations (above 1000

370 m). Figs. 6 and 7 also show that the timing of the precipitation events is roughly in agreement with the other reanalysis which assimilate ground measurements. The spatial patterns of the precipitation fields are also reproduced to some extent (see for example the highest intensities in the North-East of the domain in Figs. S2, S16, S21, and S22 in the Supplement).

The spatial resolution of ERA5-Land is too coarse to represent the dynamics of precipitation at the scale of the catchments considered in this study. Recently, convection-permitting regional climate models (CP-RCM) have shown their added value

in terms of reproduction of the highest precipitation intensities (Lucas-Picher et al., 2021). In France, Caillaud et al. (2021) show that the CP-RCM CNRM-AROME41t1, driven by CNRM-ALADINv6.2, itself driven by ERA-Interim, improves the

reproduction of large hourly and daily precipitation values, especially during the fall season. This improvement is probably due to the high spatial resolution (2.5 km grid resolution), the explicit resolution of the deep convection and a better representation of the mesoscale processes. However, the authors note that even this 2.5 km resolution might be too coarse to represent the most extreme hourly and daily intensities. Brousseau et al. (2016) show that the numerical weather prediction system AROME leads to a better reproduction of the extreme values when horizontal and vertical resolutions are increased (2.5 km and 60 levels versus 1.3 km and 90 vertical levels).

### 5.3 Added-value of ground measurements

In this study, ERA5-Land is clearly outperformed by the reanalysis for most of the metrics considered (see Fig. 3). ERA5-Land, on the contrary to the other reanalysis considered in this study, does not assimilate ground measurements and is available at a coarser spatial scale. For infrequent events (e.g. annual maxima), ERA5-Land provides smaller precipitation intensities than SPAZM and COMEPHORE at a daily or at an hourly scale (Fig. 2) which leads to a severe underestimation of the largest floods in terms of peak flow and volume (see, e.g., Fig. 7). Ground precipitation measurements assimilated in SPAZM and COMEPHORE lead to more realistic precipitation fields and provide important information about the relationship between precipitation intensities and the relief. Indeed, ERA5-Land, in the absence of further constraint about the topography, produces smooth precipitation fields (see Figs. 6 and 7), which turns out to be a severe limitation for the hydrological modelling of small mountainous catchments.

ERA5-Land is the only global-scale reanalysis considered in this study. It assimilates few in-situ measurements of precipitation (only radar data in the USA). To our knowledge, there is no global-scale reanalysis that assimilates available in-situ precipitation measurements up to date. In Europe, many recent initiatives aim at providing long reanalysis of surface variables at a high spatial and temporal resolution. For example, UERRA-Land (Soci et al., 2016) covers the period 1961-2019 at a 5.5 km spatial resolution and at a 3-hour temporal resolution (Schimanke et al., 2021). It integrates a dense precipitation observation network at a daily scale, especially in France (see Fig. 1 in Soci et al., 2016).

### 5.4 Added-value of radar information

COMEPHORE and SPAZM assimilate approximately the same number of precipitation gauges at a daily scale. In this study, intercomparisons between the results obtained with SPAZM-g and SPAZM-c inform about the added-value of the radar information concerning the subdaily dynamics of the precipitation events, since the only difference between SPAZM-g and SPAZM-c is that SPAZM-c applies the subdaily temporal structure provided by COMEPHORE when possible. Overall, there is only a marginal improvement of performances of SPAZM-c in comparison to SPAZM-g, for example for a few catchments in terms of reproduction of peak time or flood volume (see Figs. 3k and 3l). Figure 6 illustrates this difference for the flood event of 06/09/2008 in two catchments, LaGalaure@Sainte-Uze and l'Herbasse@Clérieux, for which the maximum precipitation intensity is reached earlier for SPAZM-c (cyan curve) and leads to a largest simulated peak flow and volume.

Overall, for the different metrics considered in this study, COMEPHORE leads to a better reproduction of the observed streamflow, including flood statistics (Fig. 3), this improvement being consistent over all considered catchments (Figs. 4 and

5). One striking illustration is provided in Fig. 7 where a very intense precipitation event occurred on 23/10/2013 over the catchment l'Herbasse@Clerieux according to COMEPHORE, with a maximum hourly precipitation close to 20 mm, whereas SPAZM recorded much less precipitation (100 mm versus 40 mm of cumulative precipitation amounts for COMEPHORE and SPAZM, respectively). A perfect match between observed and simulated streamflow is obtained when COMEPHORE feeds MORDOR-SD, whereas SPAZM leads to a severe underestimation of observed streamflows.

Some limitations must however be acknowledged with the use of radar data in general, and the COMEPHORE reanalysis in particular. First, there are many difficulties related to the estimation of precipitation intensities using radar signals. The quality of radar estimates is strongly dependent on the distance from the radar site, the radar signal being attenuated by various factors such as crests, precipitation meteors, etc. (Villarini and Krajewski, 2010; McRoberts and Nielsen-Gammon, 2017). As discussed in Section 5.1, COMEPHORE clearly underestimates accumulated precipitation amounts at high elevations. In this study, the hydrological models compensate for this underestimation using calibrated parameters that affect the water balance (parameters $cp$ and $exc$ for MORDOR-SD and SMASH, respectively). Second, as indicated in paragraph 2.2.4, the radar network has evolved over the period 1997-2017. Several X-band radars were installed during the period 2007-2014 but the integration of these radar measurements into the precipitation estimates was effective later, with important contributions of the radars of Mont-Maurel, Colombis, Moucherotte and la Dôle since 2016. The current radar network provides adequate coverage of most of the French Alps, except for an area near the Italian border (Haute-Tarentaise and Queyras). Third, the different methodologies applied to merge radar and in-situ measurements for the periods 1997-2006 and 2007-2017 also lead to potential inhomogeneities in the quality of the radar estimates. As a significant difference of performances of the simulated streamflow between these two periods can be suspected when COMEPHORE is used as precipitation inputs, additional analysis has been performed based on the split-sample calibration procedure. Figures S35-S36 in the Supplement show different criteria for the periods 1997-2006 and 2007-2017, for MORDOR-SD and SMASH, respectively. No major differences are noted for the $mNSE$ criteria for the streamflows $Q$ in DJFMAM, and for the $mNSE$ of the 10 largest floods and peak flows error ($\overline{PFE}$). Only a slight increase by 0.05-0.1 of the $mNSE$ criteria for the streamflow in winter and spring can be seen for about half of the catchments with MORDOR-SD, which indicates better performances for the period 2007-2017 for these catchments.

## 5.5 Problematic catchments

Overall, MORDOR-SD and SMASH hydrological models provide a fair reproduction of observed streamflows when SPAZM and COMEPHORE are used as precipitation forcings. For a few catchments, however, the evaluation metrics are clearly lower. As indicated in subsection 5.1, two of these catchments at the northwest of Grenoble have important groundwater exchanges that are not explicitly represented by MORDOR-SD and SMASH. At the event scale, the scores related to the reproduction of the ten largest floods are very low for some catchments (i.e. $mNSE < -0.3$ in Fig. 5). One of these catchments concerns L'Arve@Chamonix-Mont-Blanc, located on the eastern side of the domain. Figure 9 reports the observed and simulated streamflow of the ten largest floods for this catchment, along with the associated precipitation for the different reanalysis. Underestimation of the observed streamflow can be noticed for most of these floods, which occur mainly in summer. It is difficult to ascertain the causes of these underestimations but it can be noted that L'Arve@Chamonix-Mont-Blanc has the highest

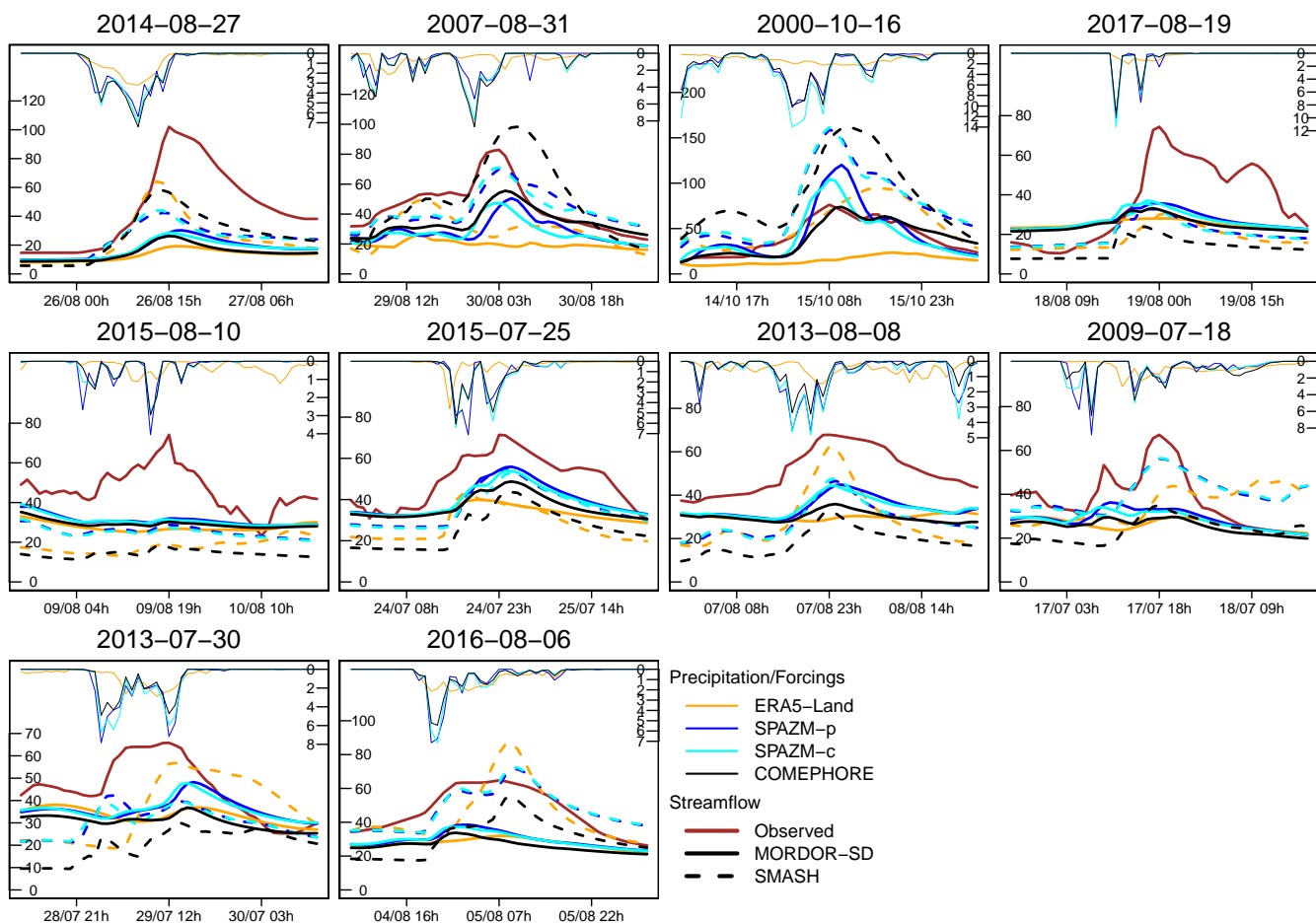

**Figure 9.** Precipitation and streamflow of the ten largest floods (highest observed peak flows) at L'Arve@Chamonix-Mont-Blanc. Time series of precipitation (thin lines) and streamflow (thick lines) for the different reanalysis.

median elevation among the catchments considered in this study. For this catchment, we can suspect that the meteorologi-
445 cal forcings are not as reliable as for the other catchments (precipitation and snow/rain partition). For another high-elevation
catchment, L'Arvan@Saint-Jean-d'Arves, peak flows are often underestimated by the simulated streamflows during the largest
floods (see Fig. S40 in the Supplement). Likely explanations are the limitations of the hydrological models to simulate the rapid
rise of surface runoff during these intense events and, similarly to L'Arve@Chamonix-Mont-Blanc, a possible underestimation
of precipitation forcings at these elevations.

**5.6   Comparison of hydrological models**

In this study, streamflow simulations are obtained using two hydrological models with different structures and modelling
choices:

- **Spatialization:** MORDOR-SD is a semi-distributed model with 12 to 14 free parameters. On the other hand, SMASH is a distributed model with 8 free parameters held constant in space (like MORDOR-SD) but where each grid cell has its representation of the hydrological processes. In this version, we do not exploit the possibility of distributing the parameters in space (Jay-Allemand et al., 2020) and opt for a more parsimonious version more easily applied to ungauged catchments using regionalisation methods.

- **Potential and actual evapotranspiration:** For both SMASH and MORDOR-SD, the potential evapotranspiration (PET) is based on the formulation provided by Oudin et al. (2005). The actual evapotranspiration (AET) is also similar for both models are they are strongly based on the structure proposed by Perrin et al. (2003) which considers that the AET depends on the saturation level of the soil moisture. However, MORDOR-SD representation of the AET is slightly more complex and also relies on the surface interception and the capillarity water storage (see section 3.1.2 in Garavaglia et al., 2017).

- **Snow module:** The two hydrological models have similar representations of snow accumulation and snow melt and the same parametric S-shaped curve for the separation into the liquid and solid parts of the snowpack. However, MORDOR-SD has more flexibility by considering additive corrections for the snowpack temperature and the rain-snow partitioning.

- **Runoff production:** In MORDOR-SD, four reservoirs control the production of the runoff (surface storage, hillslope storage, capillarity storage, and ground storage). In SMASH, the runoff is obtained from one production reservoir, two transfer reservoirs and a direct branch.

- **Routing:** The total streamflow in MORDOR-SD is the result of the surface runoff, the subsurface exfiltration, and the base flow. For SMASH, the routed streamflow is simply the sum of the runoff discharge and the upstream flow coming from a routing reservoir.

- **Parameter estimation:** The two hydrological models use different objective functions. MORDOR-SD optimizes three hydrological signatures (streamflow time series, seasonal streamflows, and flow duration curves) while SMASH is calibrated using only the streamflow time series.

To summarize, SMASH and MORDOR-SD use similar formulations of the main processes of the hydrological cycle but MORDOR-SD has additional flexibility using additional parameters and reservoirs. On the other hand, SMASH is a distributed model that might represent more adequately some dynamics at the scale of the catchment (e.g. quick surface runoff).

Figure 3 highlights some differences of performances between the two hydrological models. For the overall reproduction of streamflow (Fig. 3a-b), similar performances are obtained. SMASH has some difficulties in reproducing seasonal streamflows for many catchments (Fig. 3c-d), compared to MORDOR-SD. Interannual streamflows are often underestimated by SMASH in winter and spring and overestimated in summer and autumn (see Figure S38 in the Supplement). The criteria for flood signatures are comparable for both models, especially when COMEPHORE is used as input (Fig. 3i-k). In addition, the two hydrological models lead to the same conclusions concerning the hierarchy of the performances obtained with the different meteorological forcings.

# 6    Conclusion

This study presents an evaluation of the hydrological modelling for 55 small catchments of the Northern French Alps, focusing on the influence of the precipitation forcings. Four different precipitation products are tested. ERA5-Land assimilates satellite data and exploits a numerical weather model. SPAZM-g and SPAZM-c are two different versions of SPAZM available at a daily scale and based on precipitation gauges. Finally, COMEPHORE is a radar/gauge composite precipitation product. The semi-distributed MORDOR-SD model and the distributed SMASH model are used to simulate streamflows using the different precipitation products. Comparisons with observed streamflows highlight that:

- ERA5-Land provides the general dynamics of the precipitation events. Interestingly, ERA5-Land seems to provide a fair reproduction of annual precipitation amounts in high-elevation areas. However, it does not reproduce the spatial features of intense precipitation events at the scale of the catchments considered in this study (less than 300 km$^2$). Relationship with the relief is poorly represented and the highest intensities are underestimated. More generally, this study shows that satellite-driven reanalysis such as ERA5-Land are not likely to provide fine-scale precipitation dynamics which are necessary for the hydrological modelling of this type of small mountainous catchments.

- While fair performances are obtained with precipitation products based on a dense gauge network (i.e. SPAZM) and with a radar/gauge composite reanalysis (i.e. COMEPHORE), a clear added value of the radar information assimilated in COMEPHORE is demonstrated for the reproduction of flood events in terms of peak flows, timing, and volume. Therefore, this study shows that radar information is interesting for the hydrological modelling of small mountainous catchments despite its limitations. For example, in this study, COMEPHORE underestimates accumulated amounts at high elevations. The better performances obtained with COMEPHORE are made possible because the hydrological models compensate for these underestimations using parameters that correct the water balance.

One challenging aspect of hydrological modelling for these mountainous catchments is the adequate reproduction of the water balance at aggregated scales (e.g. annual). Precipitation inputs are highly uncertain in high-elevation areas due to the lack of direct measurements. Furthermore, groundwater exchanges or karstic sources make the representation of hydrological fluxes more difficult. However, this aspect is crucial for applications to ungauged catchments and operational applications where observed streamflows are usually absent.

**Code and data availability**

Hourly streamflow observations are available from the French HydroPortail database (http://www.hydro.eaufrance.fr/, last access: 27th April 2023). The R codes used to perform the analysis are available upon reasonable requests by directly contacting the first author.

## Author contributions

GE, MLL, and CF designed the experiments. FC and AM performed the first hydrological simulations and their evaluation. GE prepared the article with the help of all co-authors.

## Competing interests

The authors declare no conflict of interest.

## Acknowledgements

This study is part of the HYDRODEMO project which is financed by the European Union through the FEDER-POIA program and by state funds through the FNADT-CIMA program. The authors gratefully acknowledge the editor and three anonymous reviewers for their constructive suggestions.

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

## Appendix A: Disaggregation from daily to hourly precipitation and temperature using the method of fragments

### A1  Precipitation

For a day $d$, let $P_d$ denote the daily SPAZM precipitation at a pixel that we want to disaggregate. The method of fragments (Wójcik and Buishand, 2003; Breinl and Di Baldassarre, 2019) consists in using the temporal structure of another precipitation data available at a finer scale and preserving the daily amounts. Let $\tilde{P}_h$ denote the hourly precipitation for this alternative source, where $h = 1, \ldots, 24$ corresponds to the day $d$, and $\tilde{P}_d = \sum_h \tilde{P}_h$ is the corresponding daily amount. If $\tilde{P}_d$ greater than zero positive at this pixel, then the disaggregated hourly amounts $P_h$ are obtained as follows:

$$P_h = \tilde{P}_h \times \frac{P_d}{\tilde{P}_d}. \tag{A1}$$

Obviously, when $P_d = 0$, there is no precipitation to disaggregate and $P_h$ equals zero for any hour $h$ of the day. However, when $P_d$ is positive and $\tilde{P}_d$ is equal to zero, the temporal structure $\tilde{P}_d$ is absent and different solutions have been considered:

- **SPAZM-c**: COMEPHORE data are used to provide the finer precipitation data $\tilde{P}$. However, if for a day $d$ and some pixels, $\tilde{P}_d = 0$, then we look for hourly gauged precipitation data for the same day $d$ in the neighbouring region of these pixels, at a maximum distance of 100 km (see Fig. 1). If no precipitation data can be found inside this circle with a 100 km radius, $\tilde{P}$ is uniformly distributed throughout the day.

- **SPAZM-g**: This second approach is similar to SPAZM-c, except that COMEPHORE data are not used to disaggregate SPAZM precipitation, i.e. only gauged precipitation data close to the pixels are used.

### A2  Temperature

Similarly to SPAZM precipitation, daily SPAZM temperature data are disaggregated to an hourly scale using SAFRAN data as a reference. The daily ranges of SAFRAN temperatures, available at a coarser spatial scale, are corrected using the minimum and maximum daily temperature provided by the SPAZM reanalysis. For a day $d$ and a SPAZM pixel $k$, the disaggregated temperatures $T_h$ are obtained as follows:

$$T_h = (\tilde{T}_h - \tilde{T}_h^{min}) \times \frac{T_d^{max} - T_d^{min}}{\tilde{T}_d^{max} - \tilde{T}_d^{min}} + T_d^{min}, \tag{A2}$$

where $T_d^{min}$ and $T_d^{max}$ are the daily minimum and maximum SPAZM temperatures, respectively, for the pixel $k$ and the day $d$, and $\tilde{T}_d^{min}$ and $\tilde{T}_d^{max}$ are minimum and maximum SAFRAN temperatures for this day $d$ and the closest SAFRAN pixel, derived from the hourly SAFRAN temperatures $\tilde{T}_h$. The daily temperature range produced by SPAZM is thus preserved, and SAFRAN provides the subdaily temporal structure.

## Appendix B: Description of the SMASH model structure

The proposed SMASH structure for distributed modeling, based on GR lumped models (Perrin et al., 2003; Valéry, 2010) and schematized in Fig. B1, is composed of 5 reservoirs $\mathcal{S}$, $\mathcal{P}$, $\mathcal{T}_{ft}$, $\mathcal{T}_{st}$ and $\mathcal{R}$ of respective states $h_s$, $h_p$, $h_{ft}$, $h_{st}$ and $h_{lr}$ considered for simulating respectively the snow melt (CemaNeige Valéry, 2010), the production of runoff (GR4, Perrin et al., 2003), and two intermediary reservoirs (linear reservoir with leakage at exponent 5 in the differential model, Jay-Allemand et al. 2020 from GR Perrin et al. 2003) within a given cell $x$ and the routing between cells (linear reservoir). We denote the maximum capacity of the production and transfer reservoirs by $c_p$, $c_{ft}$ and $c_{st}$ respectively (snow $\mathcal{S}$ and routing $\mathcal{R}$ reservoirs have no maximum capacity).

As no solid/snow precipitation is considered as input, the initial total precipitation $P$ is divided into a liquid part named $P_l$ and a solid part $N$ using a liquid ratio computed as a function of temperature following the S-shaped parametric curve method derived from the MORDOR-SD snow module proposed by Garavaglia et al. (2017). This solid part $N$ is stored in the snow reservoir $\mathcal{S}$ whose melt $N_m$ is a function of temperature, of an estimation of the snow cover area, and of the two parameters of the snow module: $tc$, the weighting coefficient for the thermal state of the snowpack and $mc$, the melting coefficient. The evapotranpiration $E$ used as input of the model corresponds to a daily time-series of interannual potential evapotranspiration based on the formula proposed by Oudin et al. (2005), using SAFRAN temperatures, which are disaggregated at an hourly scale using a fixed subdaily distribution.

First, the net rainfall $P_n$ and the net evapotranspiration $E_n$ are obtained from the difference between $P_l + N_m$ and $E$ (i.e. $E_n = 0$ if $E_n \leq P_l + N_m$ and vice versa, see Eqs. 1 and 2 in Jay-Allemand et al., 2020). Then, the partition of $P_n$ between an infiltration part $P_s$ filling the production reservoir $\mathcal{P}$ (soil moisture accounting), and an effective rainfall $P_r = P_n - P_s$ filling the transfer reservoirs is done with a production operator. The production reservoir is then emptied from the actual evaporation $E_s$. The infiltration $P_s$ and the actual evaporation $E_s$ are functions of $P_n$ and $E_n$, respectively, $c_p$ and $h_p$ (see Eqs. 3 and 4 in Perrin et al., 2003). The effective rainfall $P_r$ is divided into 10% of direct runoff and 90% of runoff inflowing transfer part with another splitting between the two transfer reservoirs controlled by a partition parameter $\alpha$. A water exchange term $F$ depending on $exc$, which is a non-conservative operator, is applied to the direct runoff component and to the first transfer reservoir $\mathcal{T}_{ft}$ (Eq. 18 in Perrin et al., 2003). The total runoff amount $Q_t$ of a cell is therefore the sum of the flows from the direct branch $Q_d$ and the transfer reservoirs $Q_{ft}$ and $Q_{st}$ (Eqs. 20 and 22 in Perrin et al., 2003). Then, the final routed discharge amount $Q$ of a cell is the sum of $Q_t$ and the upstream flow $Q_r$ calculated using a linear routing reservoir $\mathcal{R}$ whose emptying is parameterized by $l_r$.

The numerical resolution of this ODE-based hydrological model relies on an explicit expression of its solution, approximated on the regular mesh, covering a catchment domain $\Omega$, of constant step $dx$ with a fixed time step $dt$.

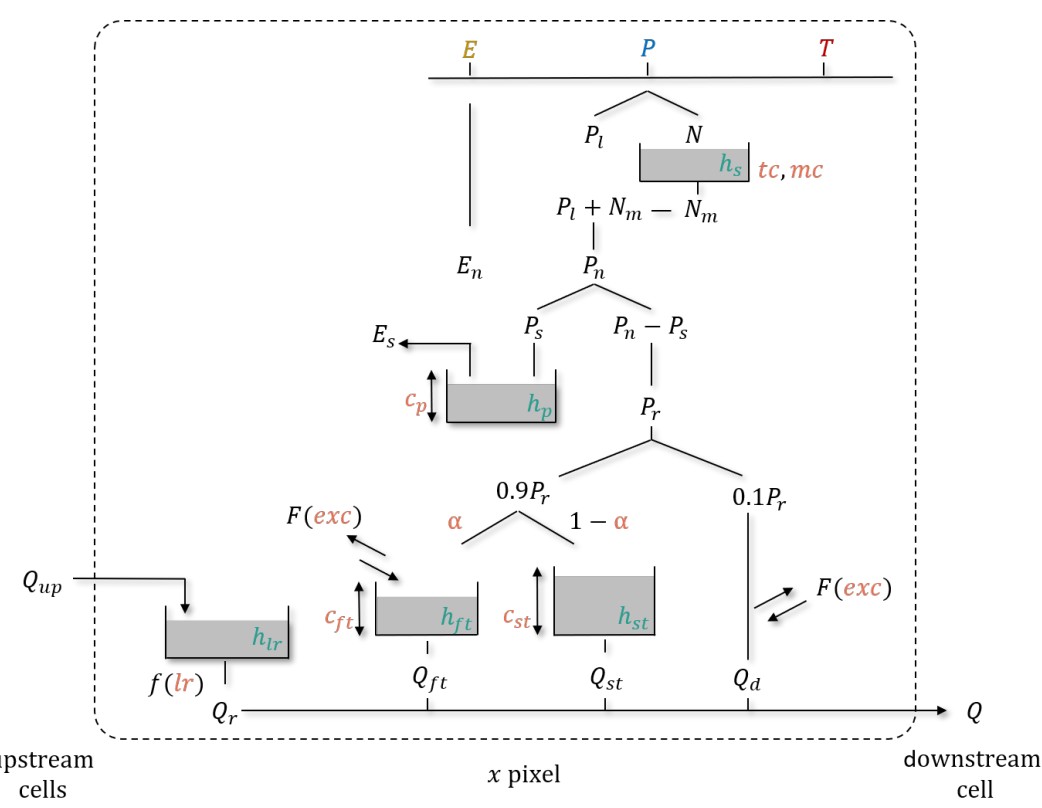

**Figure B1.** Diagram of the 8-parameter SMASH model.