# Peer review of "Evaluation of hydrological models on small mountainous catchments: impact of the meteorological forcings"

_EGUsphere, 2023_

## Referee Comment (RC2)

[referee-annotated manuscript omitted]

---

## Author Comment (AC1)

**Reviewer #1**

**RC1.1 Based on my personal reading, the presented analysis is of interest for hydrological modelling of ungauged catchments. I believe that the manuscript is well written and organized, and it deserves to be published in NHESS, after minor revisions listed below.**

We thank the reviewer for this positive feedback and the different comments that will help to improve the manuscript.

**RC1.2 Line 56: "less than 200 km2" … you mention 300km2 at line 69**

Thank you for this comment. We will replace 200 by 300 at this line. The majority of the catchments have an area smaller than 200 km$^2$ but indeed a few of them are in the range 200-300 km$^2$.

**RC1.3 Line 66. I suggest to add (in this section 2.1 or at least in the supplemental) a table with the list of the catchments with their main features (numbering, identification name, area, elevation range or average value, …). If possible, consider to report the number of each catchment in the map in figure 1, and in the explanations of results and discussion together with the catchment name. This could help the reader in identify the catchments and their main features when reading the following results and discussion section. I was sometimes lost with the different names of the catchments.**

Thank you for this suggestion that would help to follow the interpretations of the results. We will try to add numbers associated to each catchment in Figure 1, along with their names on a right panel. It is also interesting to add a table with the features of the catchments and it will be added in the manuscript.

**RC1.4 Line 225. In all the equations in section 4.1, are you using absolute differences?**

In all the equations in Section 4.1, we are using absolute differences as opposed to square differences in order to avoid inflated influences of the largest values. For some criteria (quantile relative error, volume error), these absolute differences are divided by the observed criteria and are in the end relative differences.

**RC1.5 Line 251-258. You define some errors (let's say E, where E = PFE, TPE, VE) using absolute differences, then you transform them as 1- E. I have to comments here: i) why not just considering E with sign, for having a measure of the direction of the error (under- or overestimation)?; ii) I found confusing the use of both E and 1-E in the explanations of the results (for example, lines 279-289). I suggest to use just one.**

Thank you for this comment. Concerning the first comment i), the idea was to obtain criteria that are all positively oriented, as a mixture of positively and negatively oriented criteria can also be confusing. However, considering the second comment ii), we understand that the explanations are not clear with the interpretations of both raw and transformed criteria. We propose to keep only the raw criteria in the revised manuscript and indicate clearly if they are positively or negatively oriented in Figure 3.

**RC1.6 Line 270-275. i) For better compare the results, I suggest to use same y-axis limits for the plots referring to the same type of index. For example, for mNSE in panels a-d, for QRE in panels e-h, … ii) maybe a comment is needed about SMASH model in panel d), showing huge range of mNSE compared to MORDOR-SD.**

We thank the reviewer for these constructive comments. Following comment i), we will modify figure 3 to have the same y-axis limits for similar criteria. We agree with the reviewer that it will help the comparison of the performances for different hydrological signatures. Concerning comment ii), we comment on the differences obtained with SMASH later in the discussion, at lines 449-452, but we agree that a comment could be added at this stage of the manuscript.

**RC1.7 Line 439: You mention some problems in high-elevation catchments, and (line 468) that COMEPHORE underestimates precipitation in high elevation areas. Consider to add a plot of mNSE vs elevation, for example as panels in figure4, to synthesis the maps and made more evident (if there is) a relation of mNSE with elevation.**

Thank you for this suggestion.  Please note that even if COMEPHORE underestimates precipitation in high elevation areas, this does not translate into weaker mNSE values since these underestimations can be compensated by applying correction parameters, as explained at  l. 472-474 of the manuscript. The figure R1 below shows the relationships between the different mNSE criteria and the elevation. There are no clear positive or negative relationships. In winter and spring, mNSE(Q) seems to exhibit a positive trend due to large variations of the criteria for low elevations. In summer and spring, the scatterplot seems to indicate a negative trend for elevation greater than 1000 m but this impression is mainly due to weak mNSE values obtained with SMASH. For the mNSE applied to the 10 largest floods (Fig. R1c), it is clearly highlighted that the floods are not well reproduced for some catchments located at high elevations, e.g. for the catchment L'Arve@Chamonix-Mont-Blanc. As these results do not seem to add more evidence of these relationships, they will not be included in the manuscript.

[Figure]

Figure R1. mNSE criteria as a function of the median elevation of the catchments, for the four different precipitation reanalysis and the two hydrological models. (a) *mNSE* of the streamflows *Q* in winter and spring. (b) *mNSE* of the streamflows *Q* in summer and autumn. (c) mNSE for the 10 largest floods.

**RC1.8 Line 457: "small mountainous" maybe is "small catchments".**

Thank you for noticing this missing word, this will be corrected.

**RC1.9 Supplemental, Figure S36-37. Same color scales in the panels could help in the comparisons.**

Thank you for this suggestion, we agree that it would help the comparisons between these two figures and this will be modified.

**RC1.10 Supplemental, line 16. "than with COMEPHORE" is maybe "MORDOR-SD".**

Thank you very much for your careful reading of the supplements and for noticing this mistake, this will be corrected.

---

## Author Comment (AC3)

**Reviewer #2**

**RC2.1 This very well written paper addresses the important question of how to model streamflow in small catchments with different precipitation products.**

We thank the reviewer for this overall positive opinion on this study.

**RC2.2 Overall, the paper is a bit limited in terms of references to existing literature on the question of how input resolution interacts with model performance.**

It is true that many papers have investigated the impact of input resolution for hydrological modeling, usually based on dense gauge networks (Dong et al., 2005; Meselhe et al., 2009; Bárdossy and Das, 2018; Zeng et al., 2018). For example, Xu et al. (2013) show that the increase in gauge network density can improve the performance of the hydrological model. For a large basin in China of 94,660 km$^2$, the sensitivity of the performances to the gauge density reaches a threshold when the number of gauges is high (greater than 100 gauges for this catchment). Similarly, Emmanuel et al. (2017) show that higher spatial resolution of rainfall leads to better hydrological model performance for 25 catchments with areas ranging from 42 km² to 1,855 km². Huang et al. (2019) also find that a higher temporal resolution of rainfall improves the model performance if the station density is high and that an increase in spatial resolution does not improve significantly the performance of the hydrological model for four German catchments with areas ranging from 417 km² to 1300 km$^2$. Using synthetic rainfall fields, Zhu et al. (2018) shows that the spatial resolution of precipitation has a larger impact than the temporal resolution for the simulation of floods but also depends strongly on initial soil conditions. Compared to the existing literature, a major difference in our study is that only small catchments (< 300 km$^2$) are selected and, in addition, which are located in mountainous areas where it is not known that precipitation measurements are more scarce and uncertain, particularly at high elevations. The interaction between input resolution and model performance is thus exacerbated in our study, compared to studies where catchments are often located in plains with larger areas (typically up to 1,000 km$^2$). A few papers studying the impact of rainfall resolution are focused on small urbanized catchments (Cristiano et al., 2016) or small lowland catchments (Terink et al., 2018; Hohmann et al., 2021). These references will be added to the introduction.

**RC2.3 Also, it does not discuss what different strategies actually exist to infer the hydro model meteorology at the appropriate resolution, from a meteorological product that has a different resolution and a different model topography.**

Downscaling methods and conditional simulation can be applied to obtain meteorological forcings at the appropriate resolution for the hydrological model. These approaches can be applied to disaggregate precipitation data temporally (Parkes et al., 2012, Breinl and Di Baldassarre, 2019) and/or spatially (Bárdossy and Pegram, 2016) which is also a strategy applied in our study. Some references will be added to the manuscript in Section 2.2. An important challenge for the disaggregation is the need of a

meteorological product at a fine resolution to establish the relationship between large-scale and small-scale meteorological statistics.

**RC2.4 It appears to me that an essential modelling choice is missing: how to combine the meteo product with the model? There is one option presented per hydro model but we do not know if this is a heuristic choice or the best option or what the literature says about this. In general, I do e.g. not think that retaining simply the meteo pixels within a catchment is the best option (but perhaps it is for rainfall?).**

In our knowledge, taking the average of the meteorological inputs over the pixels covering the catchment in order to estimate areal meteorological forcings is the standard approach when lumped hydrological models are applied. There exist alternative approaches when meteorological data has a coarse resolution but there are not relevant here since the reanalysis considered in this study have a fine resolution.

**RC2.5 There is a short discussion on the absence of a precip gradient for one product but perhaps it would be good to have a more systematic discussion of how to create the hydro model meteo based on the input meteo.**

Thank you for this comment. In this study, it is true that we mainly discuss the relevance of the meteorological products for hydrological modelling. How the different hydrological models process the meteorological products is a different point. In our case, MORDOR-SD assumes that precipitation and temperature have a linear relationship with the elevation, parameterized with orographic gradients parameterized by the parameters gpz and gtz, respectively. There are used to provide precipitation and temperature data for different elevation bands. SMASH is a distributed model and directly takes as inputs what is provided by the meteorological data. These aspects are really model specific, along with the choice of the parameters (fixed/inferred), the processes which are represented, the data used to estimate the free parameters. In this context, it seems difficult to provide a systematic discussion on these aspects.

**RC2.6 Also, I strongly recommend to make much clearer what take-aways are relevant beyond the studied catchments and how new they are (at the moment, there are two take-aways which do not reveal completely new insights).**

It is true that some important and general messages could be provided in the conclusions. A first important message is that ERA5-Land or satellite-driven reanalysis are not likely to provide fine-scale precipitation dynamics which are necessary for the hydrological modelling of this type of small mountainous catchments. While the limitations of ERA-Land have been discussed by a few papers (l. 209-211 of the manuscript), the implications of these limitations for hydrological applications are absent from the literature in our knowledge.

The other key message concerns the added-value of radar measurements for the hydrological modelling of these small mountainous catchments. In our case, the underestimation of annual precipitation amounts at high elevations was a known deficiency of the COMEPHORE reanalysis. Our study reveals that despite this limitation,

COMEPHORE provides better performances for some events because it provides additional information concerning the spatial extent of precipitation fields. However, this key message is really specific to this reanalysis and for the region of the study, since COMEPHORE strongly depends on the availability of radar measurements (l. 410-416). Beyond the studied catchments, a general message is that radar measurements are interesting to incorporate for the hydrological modelling of small mountainous catchments despite their limitations (radar signal attenuation by precipitation or beam blockage). While it has been shown for some flood events (Delrieu et al., 2005; Borga et al., 2007), our study provides a comprehensive evaluation of the added-value of radar data in this context, for many catchments and with two different hydrological models. The main conclusions will be rephrased in the revised version of the manuscript.

**RC2.7 What does all this for the modelling of ungauged catchments, for which no correction parameters can be calibrated? Would be cool to have some input on this question.**

Thank you for this comment. This is a very important and difficult question. In this study, correction parameters are applied mainly to balance amounts of water at aggregated scales. Therefore, it can correct different aspects that affect the water balance: lack of precipitation or precipitation excess, groundwater exchanges, impact of the karst. As such, these parameters are very difficult to regionalize since they are related to different characteristics specific to each catchment and/or to the meteorological forcings for which we have incomplete and uncertain information. This issue remains the main unresolved challenge for the application of hydrological models (conceptual models or other types) to ungauged catchments, before the representation of the hydrological system.

**RC2.8 From the abstract, the actual innovation is not clear, it seems like another paper on a previously often studied topic; what is small?**

We will precise that "small" refers to catchments with an area smaller than 300 km$^2$ in the abstract. We are not aware of similar studies comparing very different precipitation reanalysis for the hydrological modelling of many small mountainous catchments. Studies which apply hydrological models to small catchments are usually restricted to a few lowland or urban catchments, and usually consider meteorological forcings based on gauged data only (see response to comment #RC2.2).

**RC2.9 Very few references at the start of the intro, little reference to extensive literature on the role of spatial resolution of input (rainfall) forcing on the quality of hydrological simulations.**

See our response to the comment #RC2.2.

**RC2.10 From the methods, I understand that the spatial input product is aggregated to the hydro model by taking the meteo values per pixel: is this a good strategy given that the meteo product does not represent the true topography of the catchment ? In particular for the model that uses elevation bands? How are the pixel of the meteo product matched with the pixels of the hydro model for the other model?**

See our response to the comment #RC2.5.

**RC2.11 From the methods it seems like the two models are not calibrated with the same criterion, why? Is this a good idea? does this impact the analysis beyond what is mentioned in the paper?**

We agree that the choice of different criteria for the optimization of the two hydrological models impacts potentially the modelled streamflows and the corresponding results. This was made in this study to have two different hydrological models applied with the choices made by the developers of the two models (the choice of the parameters (fixed/inferred), the processes which are represented, the data used to estimate the free parameters, the input processing, etc.

**RC2.12 The paper would benefit from a concise summary of how the two models differ in terms of process representation? How do they estimate evaporation? Evaporation is almost not mentioned in the entire paper? But it should have a key influence on the representation of the water balance?**

We agree that the description of how each model estimates the evapotranspiration is missing form the current manuscript and that it should be included. For MORDOR-SD, potential evapotranspiration (PET) is estimated using the formula provided by Oudin et al. (2005). The actual evapotranspiration is estimated according to the PET and different model states. SMASH has a similar formulation.

**RC2.13 Section 3: how do the model topographies of the meteo products differ?**

See our response to the comment #RC2.5.

**RC2.14 Section 4: do I understand correctly that the models are calibrated with criteria that are discussed in the sections dedicated to the models and that other criteria are used to evaluate the performance? Or are the criteria in the model section not relevant?**

Yes, this is correct. Different criteria are used to calibrate the hydrological models and to evaluate their performance. The set of criteria used to calibrate the models is limited and has a primary objective to provide constrained estimates of the parameters and avoid parameter equifinality. The set of criteria used to evaluate their performances is a lot richer in order to obtain a comprehensive assessment of the performances.

**RC2.15 Are the models calibrated with each meteo product?**

Yes, this is correct. This will be clarified in Section 4.2.

**RC2.16 And why is the error on the floods not simply computed as a square-error, is NSE appropriate for this kind of signal? Are the values comparable to those of an entire year?**

Mean square errors depend on the magnitude of the streamflows, which can be very different from one catchment to another. The NSE is comparable to other criteria (R square, mean relative errors) and the aim here is to intercompare their values considering different meteorological forcings or hydrological models.

**RC2.17 Results: are the mNSE values a priori comparable for the different catchments? Since we do not see any streamflow time series and do not know if there are differences in the regimes, it is hard to judge.**

We agree with the reviewer that the mNSE values are not comparable for the different catchments. Here, we want to intercompare the mNSE values with different meteorological forcings or hydrological models and see if systematic differences can be observed.

References

- Bárdossy, A., et T. Das. « Influence of Rainfall Observation Network on Model Calibration and Application ». *Hydrology and Earth System Sciences* 12, n° 1 (25 janvier 2008): 77-89. https://doi.org/10.5194/hess-12-77-2008.
- Bárdossy, András, and Geoffrey G. S. Pegram. 2016. "Space-Time Conditional Disaggregation of Precipitation at High Resolution via Simulation." Water Resources Research 52 (2): 920–37. https://doi.org/10.1002/2015WR018037.
- Borga, Marco, Paolo Boscolo, Francesco Zanon, and Marco Sangati. 2007. "Hydrometeorological Analysis of the 29 August 2003 Flash Flood in the Eastern Italian Alps." *Journal of Hydrometeorology* 8 (5): 1049–67. https://doi.org/10.1175/JHM593.1.
- Cristiano, Elena, Marie-claire ten Veldhuis, Daniel B. Wright, James A. Smith, and Nick van de Giesen. 2019. "The Influence of Rainfall and Catchment Critical Scales on Urban Hydrological Response Sensitivity." *Water Resources Research* 55 (4): 3375–90. https://doi.org/10.1029/2018WR024143.
- Delrieu, Guy, John Nicol, Eddy Yates, Pierre-Emmanuel Kirstetter, Jean-Dominique Creutin, Sandrine Anquetin, Charles Obled, et al. 2005. "The Catastrophic Flash-Flood Event of 8–9 September 2002 in the Gard Region, France: A First Case Study for the Cévennes–Vivarais Mediterranean Hydrometeorological Observatory." *Journal of Hydrometeorology* 6 (1): 34–52. https://doi.org/10.1175/JHM-400.1.
- Dong, Xiaohua, C. Marjolein Dohmen-Janssen, et Martijn J. Booij. « Appropriate Spatial Sampling of Rainfall or Flow Simulation/Echantillonnage Spatial de la Pluie Approprié pour la Simulation D'écoulements ». *Hydrological Sciences Journal* 50, n° 2 (1 avril 2005): null-297. https://doi.org/10.1623/hysj.50.2.279.61801.
- Emmanuel, I., O. Payrastre, H. Andrieu, et F. Zuber. « A Method for Assessing the Influence of Rainfall Spatial Variability on Hydrograph Modeling. First Case Study in the Cevennes Region, Southern France ». *Journal of Hydrology* 555 (1 décembre 2017): 314-22. https://doi.org/10.1016/j.jhydrol.2017.10.011.
- Fu, Suhua, Torben O. Sonnenborg, Karsten H. Jensen, et Xin He. « Impact of Precipitation Spatial Resolution on the Hydrological Response of an Integrated Distributed Water Resources Model ». *Vadose Zone Journal* 10, n° 1 (2011): 25-36. https://doi.org/10.2136/vzj2009.0186.
- Hohmann, Clara, Gottfried Kirchengast, Sungmin O, Wolfgang Rieger, and Ulrich Foelsche. 2021. "Small Catchment Runoff Sensitivity to Station Density and Spatial Interpolation: Hydrological Modeling of Heavy Rainfall Using a Dense Rain Gauge Network." *Water* 13 (10): 1381. https://doi.org/10.3390/w13101381.

- Huang, Yingchun, András Bárdossy, et Ke Zhang. « Sensitivity of Hydrological Models to Temporal and Spatial Resolutions of Rainfall Data ». *Hydrology and Earth System Sciences* 23, nᵒ 6 (19 juin 2019): 2647-63. https://doi.org/10.5194/hess-23-2647-2019.
- Meselhe, E. A., E. H. Habib, O. C. Oche, et S. Gautam. « Sensitivity of Conceptual and Physically Based Hydrologic Models to Temporal and Spatial Rainfall Sampling ». *Journal of Hydrologic Engineering* 14, nᵒ 7 (1 juillet 2009): 711-20. https://doi.org/10.1061/(ASCE)1084-0699(2009)14:7(711).
- Oudin, Ludovic, Frédéric Hervieu, Claude Michel, Charles Perrin, Vazken Andréassian, François Anctil, and Cécile Loumagne. 2005. "Which Potential Evapotranspiration Input for a Lumped Rainfall–Runoff Model?: Part 2—Towards a Simple and Efficient Potential Evapotranspiration Model for Rainfall–Runoff Modelling." *Journal of Hydrology* 303 (1): 290–306. https://doi.org/10.1016/j.jhydrol.2004.08.026.
- Parkes, B. L., F. Wetterhall, F. Pappenberger, Y. He, B. D. Malamud, and H. L. Cloke. 2012. "Assessment of a 1-Hour Gridded Precipitation Dataset to Drive a Hydrological Model: A Case Study of the Summer 2007 Floods in the Upper Severn, UK." Hydrology Research 44 (1): 89–105. https://doi.org/10.2166/nh.2011.025.
- Terink, Wilco, Hidde Leijnse, Gé van den Eertwegh, and Remko Uijlenhoet. 2018. "Spatial Resolutions in Areal Rainfall Estimation and Their Impact on Hydrological Simulations of a Lowland Catchment." *Journal of Hydrology* 563 (August): 319–35. https://doi.org/10.1016/j.jhydrol.2018.05.045.
- Xu, Hongliang, Chong-Yu Xu, Hua Chen, Zengxin Zhang, et Lu Li. « Assessing the Influence of Rain Gauge Density and Distribution on Hydrological Model Performance in a Humid Region of China ». *Journal of Hydrology* 505 (15 novembre 2013): 1-12. https://doi.org/10.1016/j.jhydrol.2013.09.004.
- Zeng, Qiang, Hua Chen, Chong-Yu Xu, Meng-Xuan Jie, Jie Chen, Sheng-Lian Guo, et Jie Liu. « The Effect of Rain Gauge Density and Distribution on Runoff Simulation Using a Lumped Hydrological Modelling Approach ». *Journal of Hydrology* 563 (2018): 106-22. https://doi.org/10.1016/j.jhydrol.2018.05.058.
- Zhu, Zhihua, Daniel B. Wright, and Guo Yu. 2018. "The Impact of Rainfall Space-Time Structure in Flood Frequency Analysis." *Water Resources Research* 54 (11): 8983–98. https://doi.org/10.1029/2018WR023550.

---

## Author Comment (AC4)

**Reviewer #3**

**RC3.1 The manuscript by Evin et al., assess the performance of different precipitation products in 55 mountainous basins in France, with a specific focus on flood. They conclude that radar measurements are helpful to capture finer scale events leading to flooding, but mountain precipitation is not well captured with radar. This is a clear and straightforward case study that provide incremental insights on precipitation products in mountainous basin.**

We thank the reviewer for this positive feedback.

**RC3.2 I am a bit concerned with the scope of the objective to "choose the best product". This is very limiting as an objective as it has very limited application for a wider audience. I would reword this objective to something more applicable to a wider range of studies.  I suggest shifting the focus of the paper to be about the value added of radar information in mountain basins, as showcased by the analysis of the 55 basins, instead of having a primary objective to "select the best product".**

Thank you very much for this constructive comment. We agree with the reviewer that the outcomes of this study are not limited to the choice of the best precipitation reanalysis since three of them are only available in France. The primary objective (l. 57-59) will be rephrased in the revised version of the manuscript in order to match more closely one of the key messages: the added-value of radar measurements for the hydrological modelling of small mountainous catchments.

**RC3.3 In the introduction, I would like to see a more robust presentation of radar measurements for precipitation in mountain regions, specifically to how it performs with snow measurements.**

We agree than l.18-19 of the manuscript could be detailed in order to discuss the quality of radar measurements for precipitation in mountain regions. This is discussed in details by Germann et al. (2006) and by Villarini and Krajewski (2010), which indicate that the most important limitation of radar measurements in mountain regions is due to beam blockage (see their section 2.4). Another specific challenge is related to the phase change of precipitation that often occur between detection and arrival at ground level, potentially within/below radar elevation, which attenuates the signal within the melting layer, as shown by Khanal et al. (2019) in the northern French Alps.

A part of the literature dedicated to quantitative precipitation with radar remote sensing in complex terrain focus more specifically to the estimation of snowfall using ground measurements of snow accumulations (Rasmussen et al., 2003; Von Lerber et al., 2018). While unique reflectivity/rainfall Z/R relationships can sometimes be applied to obtain rainfall estimates, snow estimates using radar data requires different reflectivity/snowfall Z-S relationships for the different types of snow (dry/wet snow) and other factors (crystal type, degree of riming and aggregation, density, and terminal velocity, see Rasmussen et al., 2003; Khanal et al., 2019). As indicated by Von Lerber et al. (2018), an additional limitation of snowfall estimates using radar data is related to the

fact that Z/S relationships strongly rely on ground measurements of solid precipitation. However, these ground measurements are uncertain and are known to suffer from marked undercatchment. Please note that in our study, COMEPHORE does not have dedicated estimates of snowfall. Therefore, these technical aspects seem out of the scope of the paper.

**RC3.4 I found the most interesting part of the paper to be the discussion. Specifically, the analysis of model performance with different products is linked to process representation in the model (groundwater loss) and precipitation and elevation representation. I would like to see some more information on radar performance for snow vs. rain at higher elevations, and if that could cause some of these issues.**

Figure 8 presents the relationships between the correction parameter of the mode MORDOR-SD and the elevation. As indicated in the paper at l. 353-355, COMEPHORE seems to underestimate precipitation amounts in high-elevation catchments (cp > 1), since a correction parameter greater than one is likely to indicate an underestimation of the total precipitation. This underestimation is explained in section 3 at l. 216-218: "As COMEPHORE does not integrate any additional constraint about the effect of the relief, the vertical profiles of annual precipitation amounts are almost flat". The underestimation is not well understood but is probably related to the effect of beam blockage and to the different postprocessing steps applied to obtain the final precipitation estimates using the raingauge network. In the French Alps, Faure et al. (2019) provide some explanations for a similar radar/raingauge product from Météo-France (PANTHERE). They show that overestimations at low elevations and increasing underestimation at high elevations can be related to the altitudinal gradients of precipitation observed at ground level. We do not have specific results about the evaluation of COMEPHORE according to the precipitation phase but Faure et al. (2019) provide some evidence that the general underestimation of precipitation from COMEPHORE at high elevations has probably little to do with the type of meteors.

**RC3.5 Fig 8b also suggests that ERA is actually quite good at capturing high-elevation precipitation, which is a strength that could be mentioned in the conclusion.**

It is true that in this study, high-elevation total precipitation at an annual scale is actually well represented by ERA5-Land as indicated in Fig. 8b and we agree that it deserves to be added in the conclusion.

**RC3.6 The groundwater section could also be clearer: Do you mean water is exiting the basin as groundwater, so you have to reduce precipitation? It would be interesting to have more information on how this lack of process representation could be fixed, and what would be advantages of using a model with groundwater processes included in the study.**

Yes, we mean that it can be suspected that water is exiting the basin as groundwater so that the total precipitation amounts must be reduced to improve the water balance. However, this is a speculative interpretation, as we do not have much information about the groundwater processes of these catchments. MORDOR-SD can apply specific

parameterizations in order to include a conceptual representation of these groundwater processes. However, after preliminary tests, we opted for a simpler application of MORDOR-SD which avoids catchment-specific parameterizations. This was done to simplify the main messages, and for the sake of comparison between the two hydrological models.

References

- Faure, Dominique, Guy Delrieu, and Nicolas Gaussiat. 2019. "Impact of the Altitudinal Gradients of Precipitation on the Radar QPE Bias in the French Alps." *Atmosphere* 10 (6): 306. https://doi.org/10.3390/atmos10060306.
- Germann, Urs, Gianmario Galli, Marco Boscacci, and Martin Bolliger. 2006. "Radar Precipitation Measurement in a Mountainous Region." *Quarterly Journal of the Royal Meteorological Society* 132 (618): 1669–92. https://doi.org/10.1256/qj.05.190.
- Khanal, Anil Kumar, Guy Delrieu, Frédéric Cazenave, and Brice Boudevillain. 2019. "Radar Remote Sensing of Precipitation in High Mountains: Detection and Characterization of Melting Layer in the Grenoble Valley, French Alps." *Atmosphere* 10 (12): 784. https://doi.org/10.3390/atmos10120784.
- Rasmussen, Roy, Michael Dixon, Steve Vasiloff, Frank Hage, Shelly Knight, J. Vivekanandan, and Mei Xu. 2003. "Snow Nowcasting Using a Real-Time Correlation of Radar Reflectivity with Snow Gauge Accumulation." *Journal of Applied Meteorology and Climatology* 42 (1): 20–36. https://doi.org/10.1175/1520-0450(2003)042<0020:SNUART>2.0.CO;2.
- Villarini, Gabriele, and Witold F. Krajewski. 2010. "Review of the Different Sources of Uncertainty in Single Polarization Radar-Based Estimates of Rainfall." *Surveys in Geophysics* 31 (1): 107–29. https://doi.org/10.1007/s10712-009-9079-x.
- Von Lerber, Annakaisa, Dmitri Moisseev, David A. Marks, Walter Petersen, Ari-Matti Harri, and V. Chandrasekar. 2018. "Validation of GMI Snowfall Observations by Using a Combination of Weather Radar and Surface Measurements." *Journal of Applied Meteorology and Climatology* 57 (4): 797–820. https://doi.org/10.1175/JAMC-D-17-0176.1.

---

## Author Response (AR2)

**Reviewer #2**

We agree with this comment and we have significantly extended the section 5.6 dedicated to the comparison between the two hydrological models. We have tried to summarize the main differences between the two models for each of their components (spatialization, PET/AET, snow module, runoff production, routing). In a nutshell, SMASH and MORDOR-SD use similar formulations of the main processes of the hydrological cycle but MORDOR-SD has additional flexibility using additional parameters and reservoirs. On the other hand, SMASH is a distributed model that might represent more adequately some dynamics at the scale of the catchment (e.g. quick surface runoff).

#RC2.2 Would be nice to have an idea of mean precipitation of the catchments (Table 1), mean streamflow and mean simulated evaporation. This would help understand how important ET is compared to precipitation and if the different models assign a similar amount of water to ET (rather than runoff). Furthermore, the reader could understand if water balance closure problems (counterbalanced with cp or exc) can be related to the underestimation of ET. Without this information, we do not have a complete picture of the hydrological evaluation of meteorological forcings. While this might be a detail here, the fact that model calibration acts on ET is often overlooked in modelling studies, and it would therefore be good to mention here.

We also fully agree with this comment. It would be important to relate water balance closure problems to the different aspect of the hydrological models. However, we have one major technical issue related to the fact that this version of SMASH did not return the actual evapotranspiration (ETR) in the outputs. This would require re-running all the simulations with the current version of SMASH. Beside this technical limitation which is not insurmountable, some previous results (not shown) showed that the identification of the water balance closure problems is not an easy task. For example, it was not possible to see similar effects of cp and exc. As explained in section 5.1, the SMASH parameter exc enables water exchanges but is only applied to a part of the streamflow production (direct runoff branch) and has only an indirect effect on the water balance. Conversely, MORDOR-SD was designed to reproduce the water balance using two specific parameters (cp and cetp) which can act either on the precipitation inputs or the PET, respectively. Even if the parameter cetp is not used in this study, SMASH and MORDOR-SD are not really comparable concerning these questions because they have not been developed for the same purposes. MORDOR-SD seeks to reproduce all the components of the hydrological cycle (water balance, interannual variability, low/high

flows, etc.) whereas SMASH targets primarily the reproduction of flood events, which also explains why MORDOR-SD returns AET as an output and SMASH does not.

---

## Author Response (AR3)

Dear Editor,

Thank you very much for having accepted the paper for publication in HESS. We have made the technical corrections that were required:

(1) Table 1 has been moved in the Supplementary Information.

(2) The Appendix has also been moved in the Supplementary Information.

(3) We have reduced the size of most of the figures.

Sincerely yours,

Guillaume Evin on behalf of the authors.

Corresponding author: Guillaume Evin

IGE/ECRINS, 38402 Saint-Martin-d'Hères, France

guillaume.evin@inrae.fr, phone number: +33 4 7676-2821.